# Telomere dysfunction cooperates with epigenetic alterations to impair murine embryonic stem cell fate commitment

**Mélanie Criqui[1], Aditi Qamra[2†], Tsz Wai Chu[1†‡], Monika Sharma[3], Julissa Tsao[3], Danielle A Henry[1], Dalia Barsyte-Lovejoy[4], Cheryl H Arrowsmith[4], Neil Winegarden[3], Mathieu Lupien[2,5], Lea Harrington[1]***

[1]Institut de Recherche en Immunologie et Cancérologie (IRIC), Département de biologie moléculaire, Faculté de Médecine, Université de Montréal, Montréal, Canada; [2]Princess Margaret Cancer Centre, University Health Network, Toronto, Canada; [3]Princess Margaret Genomics Centre, Princess Margaret Cancer Centre, University Health Network, Toronto, Canada; [4]Structural Genomics Consortium, Princess Margaret Cancer Centre, University of Toronto, Department of Medical Biophysics, Toronto, Canada; [5]Department of Medical Biophysics, University of Toronto, Toronto, Canada

**\*For correspondence:**
lea.harrington@umontreal.ca

[†]These authors contributed equally to this work

**Present address:** [‡]Clinical Research Unit, Montreal Neurological Institute and Hospital, McGill University Health Centre, Montréal, Canada

**Competing interests:** The authors declare that no competing interests exist.

**Abstract** The precise relationship between epigenetic alterations and telomere dysfunction is still an extant question. Previously, we showed that eroded telomeres lead to differentiation instability in murine embryonic stem cells (mESCs) via DNA hypomethylation at pluripotency-factor promoters. Here, we uncovered that telomerase reverse transcriptase null (*Tert*^-/-^) mESCs exhibit genome-wide alterations in chromatin accessibility and gene expression during differentiation. These changes were accompanied by an increase of H3K27me3 globally, an altered chromatin landscape at the *Pou5f1/Oct4* promoter, and a refractory response to differentiation cues. Inhibition of the Polycomb Repressive Complex 2 (PRC2), an H3K27 tri-methyltransferase, exacerbated the impairment in differentiation and pluripotency gene repression in *Tert*^-/-^ mESCs but not wild-type mESCs, whereas inhibition of H3K27me3 demethylation led to a partial rescue of the *Tert*^-/-^ phenotype. These data reveal a new interdependent relationship between H3K27me3 and telomere integrity in stem cell lineage commitment that may have implications in aging and cancer.

## Introduction

Cellular processes crucial for development, tissue regeneration or cancer progression depend on the presence of cells that retain the capacity to commit to multiple lineages, called pluripotent or multi-potent cells. Murine embryonic stem cells (mESCs) are pluripotent and can form all tissues in the body, whereas multipotent cells can form numerous, but not all, tissue types. The capacity of these cells to replenish committed cell lineages deteriorates with age due to intrinsic and extrinsic factors (*Liu and Rando, 2011*; *Waterstrat and Van Zant, 2009*). For example, neuronal or melanocyte stem cell numbers significantly decline with age, whereas aging of hematopoietic stem cells (HSCs) leads to aberrant lineage specification and defects in cell fate determination (*Krauss and de Haan, 2016*; *Maslov et al., 2004*; *Molofsky et al., 2006*). Thus, stem cell aging directly impacts tissue function through perturbations in the stem cell reservoir or the function of their lineage-committed progeny.

The gradual attrition of chromosome ends, called telomeres, is observed in human cells in culture and in vivo, and is an established hallmark of aging that is often referred to as the 'telomere clock'

(*Harley et al., 1992*). Telomere erosion occurs in cells that lack sufficient activity of an enzyme called telomerase, whose catalytic core is comprised of a reverse transcriptase (TERT) and an integral RNA component (TR). Without telomerase, an inexorable decline in telomere reserves eventually leads to what is termed telomere uncapping. This uncapping elicits a DNA damage response that results in chromosome loss, rearrangements, and cell death or arrest (*de Lange, 2018*; *Lazzerini-Denchi and Sfeir, 2016*; *Muraki et al., 2012*; *Takai et al., 2003*). Even in stem cells, the retention of low levels of *TERT* expression cannot fully compensate for the telomere shortening that occurs during DNA replication. For example, although mice retain higher levels of telomerase activity in most adult tissues compared to humans, telomerase activity levels do decrease with age and lead to telomere erosion (*Flores et al., 2008*). Mice heterozygous for the genes encoding the telomerase RNA (*Terc*) or telomerase reverse transcriptase (*Tert*) are haploinsufficient for telomere maintenance and, depending on the genetic background, eventually exhibit many of the stem cell defects associated with telomerase knockout strains (*Harrington, 2012*; *Strong et al., 2011*). Similar telomere erosion occurs in human stem cells during aging, despite the persistence of low levels of telomerase activity (*Baerlocher et al., 2007*; *Lansdorp, 2008*; *Röth et al., 2003*; *Vaziri et al., 1994*). Although a causal link between short telomeres and human aging is still under investigation, shorter telomeres are associated with many human disorders that include dyskeratosis congenita, aplastic anemia, lung fibrosis, and blood cancer (*Adams et al., 2015*; *Armanios and Blackburn, 2012*; *Blackburn et al., 2015*; *Decottignies and d'Adda di Fagagna, 2011*). These disorders are marked by failures in cell differentiation (*Fu et al., 2018*; *Lo Sardo et al., 2017*), and by stem cell depletion in the blood and other tissues (*Bertuch, 2016*).

Post-translational modifications added to DNA and histones that are inherited over cell divisions, known as epigenetic modifications, are tightly regulated in stem cells and play a crucial role in regulating cell fate. Many epigenetic alterations occur with age and are considered hallmarks of the aging process (*Booth and Brunet, 2016*). For example, DNA methylation patterns are altered during aging, whereby the genome becomes largely hypomethylated with hypermethylation at some loci (*Krauss and de Haan, 2016*). One contributing mechanism may be the downregulation of DNA methyltransferase (Dnmt) activity with age (*Sun et al., 2014*). Despite the finding that DNA hypomethylation does not affect stem cell self-renewal capacity nor the expression of pluripotency genes (*Beerman and Rossi, 2015*), *Dnmt3a* knock-out mice exhibit an increase in HSC self-renewal and a predisposition to hematopoietic malignancies (*Mayle et al., 2015*). Changes in the abundance of other epigenetic modifications, such as decreased tri-methylation of histone H3 on lysine 27 (H3K27me3) is associated with and may help drive the onset of senescence (*Ito et al., 2018*; *Shah et al., 2013*). Conversely, an increase of H3K27me3 in HSCs and muscle stem cells observed in aged mice is suggested to restrict stem cell potential via a differentiation bias of older stem cells (*Brunet and Rando, 2017*). Given the role of epigenetics in stem cell differentiation and consequently in tissue maintenance, these and other findings have led to the notion that epigenetic alterations represent another age-associated clock (*Hannum et al., 2013*; *Horvath, 2013*; *Jung and Pfeifer, 2015*; *Wilson and Jones, 1983*). These age-associated changes in DNA methylation patterns may intersect a broad array of health-related issues, ranging from Alzheimer's disease to circadian rhythm disruption to lactose intolerance (*Labrie et al., 2016*; *Oh et al., 2019*; *Oh et al., 2018*; *Oh et al., 2016*).

A major question that remains unresolved is the role of histone methylation in the epigenetic regulation of cell fate in cells with dysfunctional telomeres. In mice, loss of telomere integrity affects chromatin throughout the telomeric and sub-telomeric DNA (*Benetti et al., 2008*; *Benetti et al., 2007*; *Gonzalo et al., 2006*). More recently, roles distal to the telomere have been described for the telomere shelterin subunit, Rap1, in mice and in budding yeast (*Marión et al., 2017*; *Platt et al., 2013*; *Song and Johnson, 2018*; *Vaquero-Sedas and Vega-Palas, 2019*). There are other consequences for cell fate beyond senescence, as telomere dysfunction is known to impair stem cell differentiation in numerous contexts (*Harrington and Pucci, 2018*). For example, eroded telomeres in mESCs impair their ability to fully consolidate a differentiated phenotype in response to all-trans retinoic acid (ATRA) (*Pucci et al., 2013*). Repression of the pluripotency-promoting factors *Nanog* and *Pou5f1* (also known as *Oct4*) is impaired in late passage cells with critically short telomeres upon differentiation, compared to wild-type (WT) or earlier passage telomerase-deficient (*Tert*$^{-/-}$) mESCs with longer telomeres. This impairment is associated with a reduction in DNA methylation at the *Nanog* promoter (*Pucci et al., 2013*). The impaired repression of pluripotency genes in mESCs with

dysfunctional telomeres is reversible, and telomere rescue via *Tert* reintroduction, or ectopic expression of *Dnmt3b*, elicits a partial reversal of this differentiation defect (*Pucci et al., 2013*). Although the ability of critically eroded telomeres to induce these changes is unexpected, the consequences of these epigenetic alterations upon cell fate commitment was already well known (*Kraushaar and Zhao, 2013*; *Nativio et al., 2018*; *Reddington et al., 2013*; *Smith et al., 2010*).

Since the repression of *Nanog* and *Pou5f1* during differentiation rely upon H3K27me3 deposition via the Polycomb Repressive Complex 2 (PRC2) (*Obier et al., 2015*; *Villasante et al., 2011*), and Dnmt3 and PRC2 are critical to mESC neural differentiation (*Liu et al., 2018*), we investigated whether perturbation of H3K27me3 levels, genome-wide, could further impact the cell differentiation defect associated with dysfunctional telomeres. We found a striking ability of chemical modulators of histone H3K27 methylation to specifically affect the fate of mESCs with short telomeres, with little effect on WT mESCs. We suggest that telomere status and histone methylation may be more linked than previously appreciated.

## Results

### Murine ESCs with critically short telomeres fail to consolidate a differentiated state

To address the role of telomere shortening on stem cell differentiation, we analyzed late-passage (passage 40 or greater) $Tert^{-/-}$ mESCs that possess significantly shorter telomeres and an increase in end-to-end fusions compared to WT mESCs (*Figure 1—figure supplement 1A,B*; *Liu et al., 2000*). Despite this telomere instability, and consistent with our previously reported results, there was no observable difference in population doubling time between the lines (*Pucci et al., 2013*) (data not shown). In WT and $Tert^{-/-}$ mESCs, we assessed stem cell differentiation by the withdrawal of Leukemia Inhibitory Factor (LIF) and the induction of cell differentiation with 5 µM all-trans retinoic acid (ATRA) for 6 days (6 DA; blue) (*Figure 1A*). The stability of the differentiated state, that is a refractory response to LIF re-addition, was also evaluated upon a further 6 days in LIF-containing media that lacked ATRA (6 DA + 6 DL; orange) (*Figure 1A*). Both mESC lines contained a green fluorescent protein (GFP) reporter placed under the transcriptional control of the pluripotency gene regulator *Pou5f1*, which enabled their sorting into distinct pluripotent (GFP+) or lineage-committed (GFP-) sub-populations (*Figure 1A*). As expected after induction of differentiation (6 DA), WT mESCs went from a GFP+ state (green) to a GFP- state (blue) and remained so even after LIF re-addition (orange) (*Figure 1B*). In contrast, $Tert^{-/-}$ mESCs failed to fully repress the GFP reporter when treated with ATRA, and this defect became more pronounced after re-addition of LIF-containing media (*Figure 1B*). The failure of $Tert^{-/-}$ mESCs to repress *Pou5f1*-GFP during differentiation was also reflected in higher endogenous levels of *Pou5f1* and *Nanog mRNA* (*Figure 1C,D*). GFP cell-sorting and further analysis of other pluripotency markers showed that GFP+ sorted sub-populations of $Tert^{-/-}$ mESCs retained a higher level of pluripotency gene expression after ATRA (SA), or LIF re-addition (SL) (*Figure 1—figure supplement 1C*). These results are consistent with previous reported observations in WT or $Tert^{-/-}$ mESCs that did not contain the *Pou5f1*-GFP reporter (*Pucci et al., 2013*).

We also found that gamma-irradiation up to 5 Gy did not affect the GFP status of WT mESCs grown in LIF or after differentiation, which argues against a general effect of DNA damage on differentiation commitment under these conditions (*Figure 1—figure supplement 1E–H*). In contrast, $Tert^{-/-}$ mESCs, although slightly less radio-sensitive than WT mESCs (*Figure 1—figure supplement 1E*), exhibited a further defect in differentiation commitment (*Figure 1—figure supplement 1G,H*). These results suggest that DNA damage may enhance the impact of critically eroded telomeres, but that the effect is not merely attributable to DNA damage per se.

### Compromised telomere integrity alters gene expression profiles

Differentiation into a specific cell type is highly context-dependent, and the ability of ATRA to affect gene expression during mESC differentiation in vitro has been well documented (*Simandi et al., 2010*). To further query the lineage commitment of $Tert^{-/-}$ mESCs, we examined the expression pattern of 74 lineage-specific genes and five pluripotency genes using a commercially available quantitative real-time polymerase chain reaction (qPCR) array specific to murine cell lineages (*Figure 1E*,

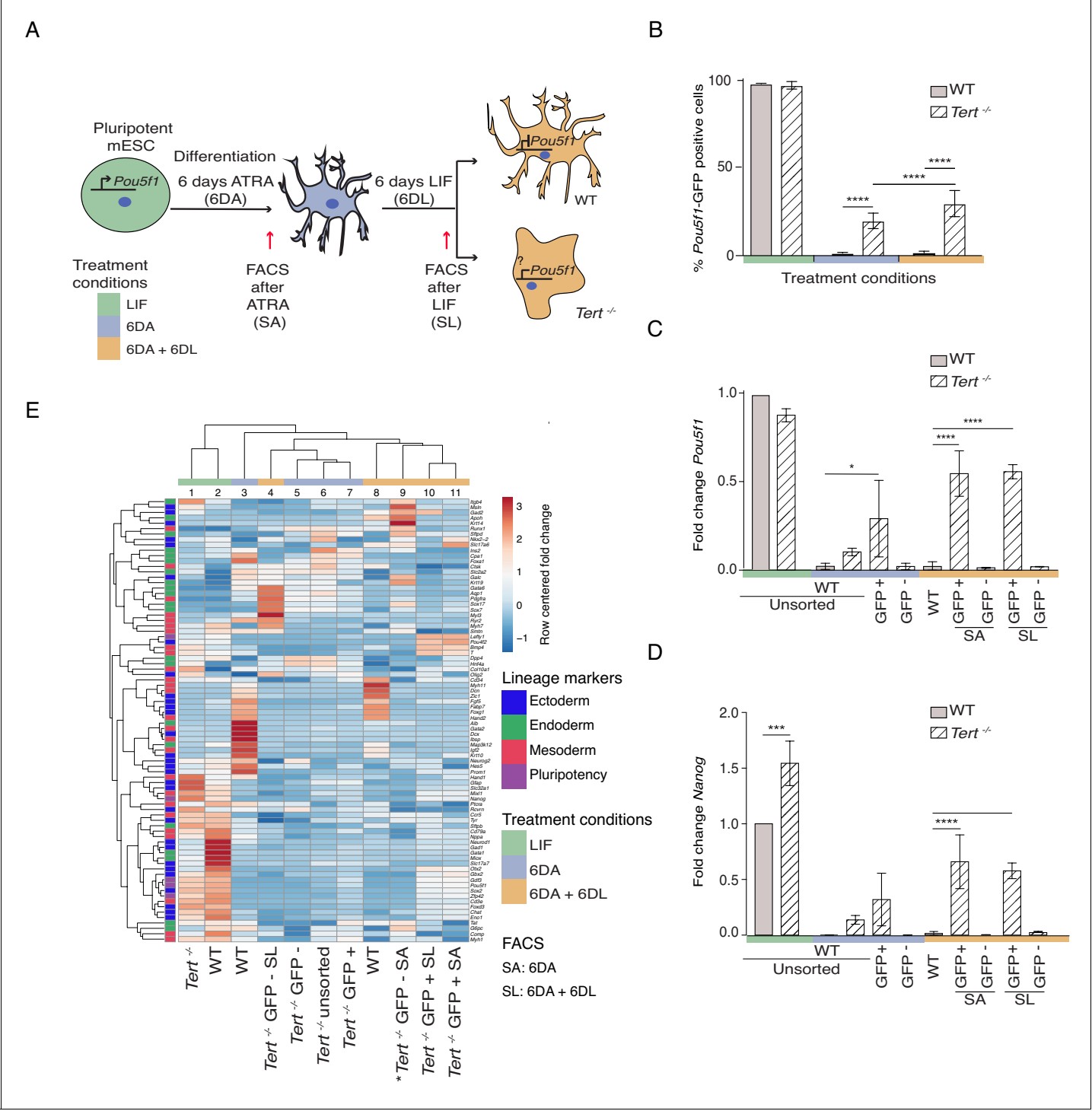

**Figure 1.** Murine ESCs with short telomeres fail to complete differentiation commitment and to suppress pluripotency gene expression. (**A**) Schematic representation of the experimental design. Wild-type (WT) and *Tert*⁻/⁻ mESCs transduced with *Pou5f1* -GFP were maintained in a pluripotent state in the presence of leukemia inhibitory factor (LIF). Murine ESCs were next induced to differentiate by all-trans retinoic acid (ATRA) treatment for 6 days, followed by re-seeding in LIF-containing media for an additional 6 days to assess their ability to remain stably differentiated (GFP-). The GFP status of mESCs was monitored by flow cytometry (FACS) and sorted samples were collected at each experimental step: (i) LIF (pluripotent), (ii) 6 days ATRA (6 DA) and (iii) 6 days ATRA followed by 6 days LIF (6 DA + 6 DL). SA: mESCs sorted after 6 DA. SL: mESCs sorted after 6 DA + 6 DL (**B**) FACS analysis of *Pou5f1*-GFP reporter expression in mESCs at each experimental step. Data are represented as mean ± SD. Data represent n = 3 biological replicates: for n = 1 and n = 2 biological replicates, there were three technical replicates; for the n = 3 biological replicate, there were two technical replicates, for a total of 8 samples. Statistical analysis was performed using two-way ANOVA, * (p<0.0332), ** (p<0.0021), *** (p<0.0002), **** (p<0.0001). Color bars

*Figure 1 continued on next page*

Figure 1 continued

underneath the x-axis represent the same treatment groups as indicated in (A). (C, D) Assessment of relative fold change in *Pou5f1* and *Nanog* expression (WT pluripotent set as 1) by RT-qPCR, normalized to five housekeeping genes: *Actb*, *B2m*, *Gapdh*, *Gusb* and *Hsp90ab1*. Data are represented as mean ± SD (n = 3). Statistical analysis was performed using ordinary one-way ANOVA, * ($p<0.0332$), ** ($p<0.0021$), *** ($p<0.0002$), **** ($p<0.0001$). Color bars underneath the x-axis represent the same treatment groups as indicated in (A). (E) Heatmap representation of relative fold change in gene expression of 77 lineage-specific markers (endoderm, mesoderm and ectoderm) and five pluripotency genes from the Qiagen Mouse Cell Lineage Identification qPCR Array (n = 3 except for one sample marked * where n = 2, see legend D). Relative fold change in gene expression was calculated using WT pluripotent mESC (2 days in LIF) as control (fold change = 1) and normalized to five housekeeping genes: *Actb*, *B2m*, *Gapdh*, *Gusb* and *Hsp90ab1*. The color scale represents the row centered fold change values (blue = lower, white = intermediate, red = higher). Euclidean distance clustering and complete linkage was applied to visualize similarity between samples. See Materials and methods for details.

The online version of this article includes the following source data and figure supplement(s) for figure 1:

**Source data 1.** Murine ESCs with short telomeres fail to complete differentiation commitment and to suppress pluripotency gene expression.
**Figure supplement 1.** Telomere status, gene expression, and gamma-irradiation response in WT and *Tert*[-/-] mESCs.

see Materials and methods). Euclidean clustering and principal component analysis of the transcriptional data revealed heterogeneity between the cell populations, however a few key observations emerged. First, as anticipated the transcriptional profiles between undifferentiated WT and *Tert*[-/-] mESCs were more alike than other differentiated cell populations (*Figure 1E*, columns 1 and 2, *Figure 1—figure supplement 1D*, circle 1). Second, in keeping with the differences observed in pluripotency gene expression profiles, WT mESCs treated with ATRA formed a distinct group relative to ATRA-treated *Tert*[-/-] mESCs (*Figure 1E*, column 3 and 5–7, *Figure 1—figure supplement 1D*, compare blue and orange triangles with circles 2, 3). Third, within the populations treated with ATRA and LIF (6 DA + 6 DL) (*Figure 1E*, columns 4, 8–11), there was a difference between *Tert*[-/-] mESCs that were sorted for GFP- or GFP+ status (*Figure 1E*, column 4, *Figure 1—figure supplement 1D*, circles 2 and 3). These data suggest there are at least two distinct sub-populations within *Tert*[-/-] mESCs treated with ATRA, or ATRA and LIF: GFP+ populations in which pluripotency genes fail to be repressed (*Figure 1—figure supplement 1C*), and GFP- populations whose transcriptional profiles differ from GFP- WT mESCs (*Figure 1E*, compare columns 3, 8 versus 4–6). In summary, the transcriptional profiles in *Tert*[-/-] mESCs after differentiation induction are consistent with the conclusion that some cells are unable to attain and maintain a differentiated state.

## Inhibition of PRC2 specifically impairs differentiation in mESCs with dysfunctional telomeres

Telomere dysfunction in mESCs results in reduced expression of the de novo DNA methyltransferases 3a and 3b (Dnmt3a/3b) (*Pucci et al., 2013*). As we demonstrated previously using an antibody that we validated for its specificity against H3K27me3, global H3K27me3 levels were also upregulated in *Tert*[-/-] mESCs (*Pucci et al., 2013*; *Figure 2—figure supplement 1A*). While DNA methylation is known to assist in the recruitment of histone methyltransferases, and H3K27me3 serves an essential role during differentiation (*Jones and Wang, 2010*; *Margueron and Reinberg, 2011*), it was unclear if the alterations in H3K27me3 levels in *Tert*[-/-] mESCs impinged directly on differentiation commitment. We thus probed the epistatic relationship between telomere dysfunction and H3K27me3 deposition using chemical inhibitors of the H3K27 methyltransferase PRC2. We tested three different inhibitors of the PRC2 complex that targeted two of its core subunits: Enhancer of zeste homolog, Ezh2 (UNC1999, GSK343) or Embryonic ectoderm development, Eed (A395) (*He et al., 2017*; *Konze et al., 2013*; *Verma et al., 2012*). UNC2400 and A395N are structurally related to UNC1999 and A395 respectively, but exhibit a reduced inhibition activity against PRC2 (*He et al., 2017*; *Konze et al., 2013*) (for further details refer to Materials and methods). These inhibitors were chosen because of the extensive biochemical and in vivo data to establish their specificity for their cognate substrates (*He et al., 2017*; *Konze et al., 2013*; *Shi et al., 2017*). Although *Ezh2 mRNA* was slightly lower in undifferentiated *Tert*[-/-] ESCs compared with WT ESCs, *Ezh2*, *Suz12*, and *Eed* continued to be expressed after differentiation at levels that did not significantly differ from WT ESCs (*Figure 2—figure supplement 1B*). The other active orthologue of *Ezh2*, *Ezh1*, was not detected in mESCs (data not shown). Thus, we presume the major target of PRC2 inhibition in this cellular context is *Ezh2* (this presumption was also tested genetically; see below).

Since these inhibitors had previously been studied primarily in human cell lines (*He et al., 2017*; *Konze et al., 2013*; *Verma et al., 2012*), we first optimized and selected the concentration of PRC2 inhibitors that did not affect the overall population doubling rate of either WT or *Tert*$^{-/-}$ pluripotent mESCs after 2 days in LIF media (data not shown). Using these optimized concentrations, we incubated mESCs with each compound throughout the differentiation assay, over a total period of 14 days (*Figure 2—figure supplement 1C*; for further details refer to Materials and methods). Western blot analysis showed that the active PRC2 inhibitors, but not their inactive analogues, reduced global H3K27me3 in WT and *Tert*$^{-/-}$ mESCs (*Figure 2A* and *Figure 2—figure supplement 1D*). Irrespective of genotype, mESCs maintained in LIF or treated with ATRA (6 DA) exhibited no significant alteration in the expression of *Pou5f1*-GFP upon treatment with PRC2 inhibitors or their inactive analogues, relative to DMSO-treated controls (*Figure 2—figure supplement 1E*). Therefore, we focused our subsequent analysis on cell populations treated for 14 days, after ATRA treatment and LIF re-exposure (6 DA + 6 DL).

We found that PRC2 inhibition led to a marked exacerbation of the differentiation defect in *Tert*$^{-/-}$ mESCs. A significant increase in the percentage of GFP+ cells was observed in *Tert*$^{-/-}$ mESCs treated with PRC2 inhibitors for 14 days, compared with the inactive analogues or DMSO-treated cells (*Figure 2B*). The expression of endogenous *Nanog mRNA* or protein (*Figure 2C*, and *Figure 2—figure supplement 1F,G*), and other pluripotency gene markers such as *Gdf3*, *Lefty1* and *Zfp42* were also further increased relative to *Tert*$^{-/-}$ mESCs treated with DMSO (*Figure 2—figure supplement 1G*). In contrast, WT mESCs exhibited no significant increase in the percentage of GFP + cells after 14 days of inhibitor relative to DMSO treatment (*Figure 2B*) and differentiation induction of WT mESCs led to a sustained downregulation of pluripotency factor expression (*Figure 2C*, *Figure 2—figure supplement 1F,G*). As a control for the possible effects of exposure time, compounds added to undifferentiated WT or *Tert*$^{-/-}$ mESCs for up to 6 days did not alter the percentage of GFP+ cells (data not shown). These results demonstrate that the differentiation consolidation of *Tert*$^{-/-}$ mESCs, but not WT mESCs, was markedly affected by PRC2 inhibition.

To further examine the impact of PRC2 inhibitors on transcription, RNA was isolated from cells treated with DMSO or inhibitors throughout differentiation, and subjected to qPCR analysis using the same cell lineage array as above. Heatmap and principal component analysis showed that PRC2 inhibition also led to an alteration in the overall gene expression profiles in *Tert*$^{-/-}$ mESCs after treatment with the PRC2 inhibitors GSK343, UNC1999 and A395 compared with their inactive analogues (UNC2400, A395N) or DMSO treatment alone (*Figure 2D*, *Figure 2—figure supplement 1H*). These data further establish that PRC2 inhibition elicited alterations in the transcriptional program that accompanied the defect in differentiation commitment.

## Inhibition of the H3K27me3 erasers Kdm6a/b partially rescues the differentiation impairment of *Tert*$^{-/-}$ mESCs

Previously, we showed that global DNA hypomethylation in *Tert*$^{-/-}$ mESCs, and the reduced hypomethylation of the *Nanog* and *Pou5f1* promoters could be partially alleviated by ectopic expression of *Dnmt3b* (*Pucci et al., 2013*). This capacity for phenotype reversibility prompted us to examine whether inhibition of the JmjC-containing protein and H3K27 demethylase Kdm6b (also known as Jmjd3) might also alleviate the differentiation impairment in *Tert*$^{-/-}$ mESCs. We first assessed *Kdm6b and Kdm6a* (UTX) *mRNA* expression, and found that *mRNA* levels were comparable in *Tert*$^{-/-}$ ESCs and WT ESCs (*Figure 3—figure supplement 1A,B*). Wild-type and *Tert*$^{-/-}$ mESCs were treated with the Kdm6a/b inhibitor GSKJ4, or its inactive analogue GSKJ5 (*Kruidenier et al., 2012*; *Li et al., 2018*), under the same experimental conditions described above (*Figure 3—figure supplement 1C*). Despite the modest and statistically insignificant impact of GSKJ4 treatment on global H3K27me3 levels (*Figure 3A* and *Figure 3—figure supplement 1D*), there was a significant reduction in the GFP+ population in *Tert*$^{-/-}$ mESCs compared with DMSO or GSKJ5 treatment, and no observable impact on WT mESCs (*Figure 3B*). Quantitative PCR gene expression and western blot analysis showed that *Pou5f1*, *Nanog mRNA* and protein, and other endogenous pluripotency markers were modestly repressed in GSKJ4-treated *Tert*$^{-/-}$ mESCs relative to DMSO-treated cells (*Figure 3C*, and *Figure 3—figure supplement 1E,F*). Heatmap and principal component analysis revealed that GSKJ4 treatment altered the overall transcriptional profile of *Tert*$^{-/-}$ mESCs (*Figure 3D*, *Figure 3—figure supplement 1G*), whereas GSKJ5 and DMSO-treated cells were more similar to each other. GSKJ4 also has a lower affinity for other JmjC-containing proteins, including

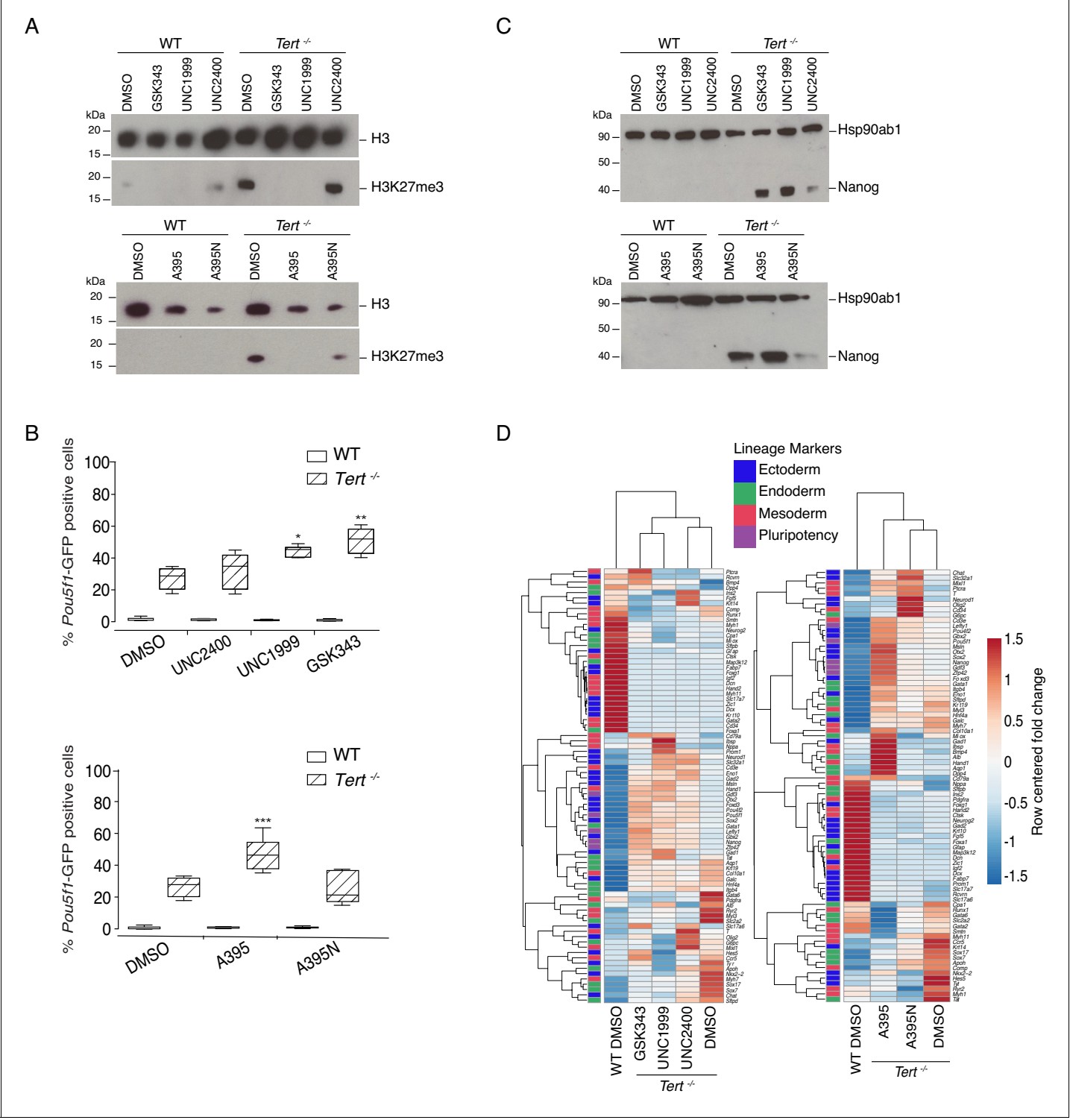

**Figure 2.** Murine ESCs with short telomeres exhibit altered H3K27me3 levels and incomplete differentiation that is exacerbated by PRC2 inhibition. (**A**) Western blot analysis of H3K27me3 levels in WT and *Tert*[-/-] mESCs, in the presence of active PRC2 inhibitors (GSK343, UNC1999 and A395), their corresponding controls (UNC2400 and A395N), or the vehicle-only control DMSO. H3 was used as a loading control. Representative blot from n = 3 biological replicates (see *Figure 2—figure supplement 1D* for quantification of additional representative blots). (**B**) FACS analysis of *Pou5f1*-GFP reporter expression in mESCs. Data are represented as mean ± SD. Data represent n = 3 biological replicates: for n = 1 and n = 2 biological replicates, there were three technical replicates; for the n = 3 biological replicate, there were two technical replicates, for a total of 8 samples. Statistical analysis was performed using two-way ANOVA, * (p<0.0332), ** (p<0.0021), *** (p<0.0002), **** (p<0.0001). Graphs were generated using the online tool

*Figure 2 continued on next page*

*Figure 2 continued*

BoxPlotR (http://shiny.chemgrid.org/boxplotr/). (**C**) Western blot analysis of Nanog protein expression. Hsp90ab1 was used as a loading control. Representative blot from n = 3 biological replicates (see *Figure 2—figure supplement 1F*). (**D**) Heatmap representation of relative fold change in gene expression of 77 lineage-specific markers (endoderm, mesoderm and ectoderm) and five pluripotency genes from the Qiagen Mouse Cell Lineage Identification qPCR Array (n = 3). Gene expression analysis was performed on 6 DA + 6 DL mESCs treated with active PRC2 inhibitors (GSK343, UNC1999 and A395), the inactive control compounds (UNC2400 and A395N) or DMSO. Relative fold change in gene expression was calculated using *Tert*$^{-/-}$ DMSO as control (fold change = 1) and normalized to five housekeeping genes: *Actb*, *B2m*, *Gapdh*, *Gusb* and *Hsp90ab1*. The color scale represents the row centered fold change values (blue = lower, white = intermediate, red = higher). Euclidean distance clustering and complete linkage was applied to visualize similarities between samples.

The online version of this article includes the following source data and figure supplement(s) for figure 2:

**Source data 1.** Murine ESCs with short telomeres exhibit altered H3K27me3 levels and incomplete differentiation that is exacerbated by PRC2 inhibition.
**Figure supplement 1.** The impact of PRC2 inhibitors on H3K27me3, gene expression, and differentiation of WT and *Tert*$^{-/-}$ mESCs.

the H3K4 di- and tri-demethylase encoded by *Kdm5b* (*Heinemann et al., 2014*). These data suggest that the differentiation instability of *Tert*$^{-/-}$ mESCs can be partially rescued by inhibition of the molecular targets of GSKJ4.

We found that the differentiation instability of *Tert*$^{-/-}$ mESCs could be partially rescued by inhibition of H3K27 de-methylation. This finding would be expected if, as we observed, PRC2 inhibition exacerbated the differentiation stability of *Tert*$^{-/-}$ mESCs. In addition, the opposing roles of these gene products on H3K27me3 are consistent with previously published effects of PRC2 and Kdm6a/b inhibitors on human cells (*Konze et al., 2013*; *Kruidenier et al., 2012*). To further genetically test these findings, we performed a CRISPR-Cas9 targeting of *Ezh2* or *Kdm6b* in mESCs (*Figure 3—figure supplement 1H,I*). It is known that clonal WT mESCs deleted for *Ezh2* or *Kdm6b* exhibit marked differentiation defects (*Chamberlain et al., 2008*; *Lavarone et al., 2019*; *Sen et al., 2008*). Thus, to improve the ability to detect differences between WT and *Tert*$^{-/-}$ mESCs, we disrupted *Ezh2* or *Kdm6b* using CRISPR-Cas9 and analysed the bulk, non-clonal population. In these heterogeneous populations, in which a fraction of cells were disrupted for *Ezh2* or *Kdm6b*, H3K27me3 levels were modestly altered (*Figure 3—figure supplement 1I, J, K*). Neither gene-targetted population exhibited a difference in the differentiation capacity of WT mESCs, but in *Tert*$^{-/-}$ mESCs, *Ezh2* knockdown led to a statistically significant impairment in differentiation (*Figure 3—figure supplement 1L*). Conversely, *Kdm6b* knockdown partially alleviated this defect (*Figure 3—figure supplement 1L*). These genetic data further bolster our findings using chemical inhibitors of Ezh2 and Kdm6b.

## Telomere dysfunction remodels the chromatin accessibility landscape of differentiated cells toward a pluripotent-like state

The differentiation impairment and transcriptional alterations observed in *Tert*$^{-/-}$ mESCs begets the question of whether there is a more global alteration of the chromatin landscape in mESCs with telomere dysfunction. It is known that global epigenetic remodelling is required for embryonic stem cell differentiation, whereby a generally open hyperdynamic chromatin state transitions to a more stable and closed state (*Chen and Dent, 2014*; *Gaspar-Maia et al., 2011*; *Wang et al., 2015*; *Zhu et al., 2013*). To assess how the chromatin accessibility landscape was remodelled during differentiation in WT and *Tert*$^{-/-}$ mESCs, we performed ATAC (Assay for Transposase-Accessible Chromatin)-seq on cells isolated at various steps throughout the differentiation treatment, with or without compounds that modulated PRC2 or histone demethylase activity (see Materials and methods for details).

Spearman correlation between ATAC-seq profiles of mESC populations highlighted a distinct chromatin accessibility landscape in differentiated *Tert*$^{-/-}$ mESCs compared to WT mESCs. In differentiated WT mESCs, there was a high degree of similarity between all samples with or without treatment with PRC2 or Kdm6a/b inhibitors (*Figure 4A*, samples 1–9). In contrast, undifferentiated WT mESCs more closely resembled *Tert*$^{-/-}$ mESCs (*Figure 4A*, samples 10–24). Of these latter *Tert*$^{-/-}$ mESC chromatin accessibility profiles, the GSK343-treated cells were most similar to WT undifferentiated mESCs (*Figure 4A*, samples 14–15). This similarity of the ATAC-seq profile to undifferentiated *Tert*$^{-/-}$ mESCs was also evident with PRC2 or Kdm6a/b inhibitor treatment (*Figure 4A*, samples 16–24). Consistent with the principal component analysis (*Figure 4—figure supplement 1C*), *Tert*$^{-/-}$ mESCs treated with ATRA alone also differed from WT mESCs treated with ATRA (*Figure 4A*, compare samples 1 versus 10–12), and clustered with GFP- *Tert*$^{-/-}$ mESCs treated with ATRA + LIF

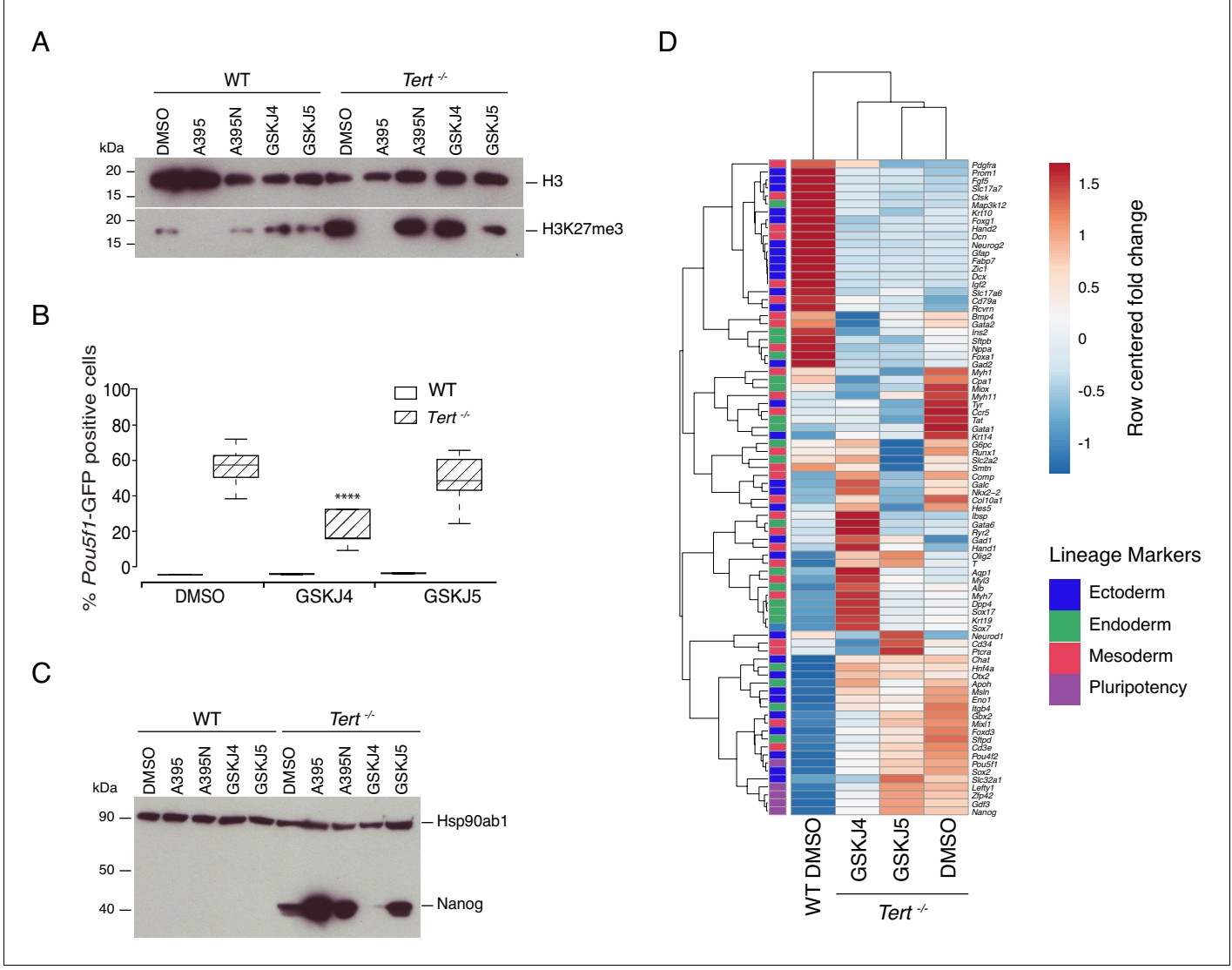

**Figure 3.** Inhibition of Kdm6a/b histone demethyltransferase activity partially rescues cell fate commitment. (A, C) Western blot analysis of H3K27me3 or Nanog protein levels in WT and *Tert*[-/-] mESCs, in the presence of DMSO only, GSKJ4 or GSKJ5. For clarity, no image cropping was performed between lanes and the blot includes an independent replicate of the A395/A395N treatments shown in *Figure 2*. Hsp90ab1 and H3 are used as loading control for Nanog and H3K27me3 respectively. Representative blots from n = 3 biological replicates. Quantification of western blots is shown in *Figure 3—figure supplement 1D,G*. (B) FACS analysis of *Pou5f1*-GFP reporter expression in mESCs. Data are represented as mean ± SD (n = 3 biological replicates each with three technical replicates). Statistical analysis was performed using two-way ANOVA, * ($p<0.0332$), ** ($p<0.0021$), *** ($p<0.0002$), **** ($p<0.0001$). (D) Heatmap representation of relative fold change in gene expression of 77 lineage-specific markers (endoderm, mesoderm and ectoderm) and five pluripotency genes from the Qiagen Mouse Cell Lineage Identification qPCR Array (n = 3 biological replicates). Gene expression analysis was performed on 6 DA + 6 DL mESCs treated with the Kdm6a/b inhibitor (GSKJ4), the inactive molecule (GSKJ5) or DMSO control. Relative fold change in gene expression was calculated using *Tert*[-/-] DMSO as control (fold change = 1) and normalized to five housekeeping genes: *Actb*, *B2m*, *Gapdh*, *Gusb* and *Hsp90ab1*. The color scale represents the row centered fold change values (blue = lower, white = intermediate, red = higher). Euclidean distance clustering and complete linkage was applied to visualize similarity between samples.

The online version of this article includes the following source data and figure supplement(s) for figure 3:

**Source data 1.** Inhibition of Kdm6a/b demethylase activity partially rescues cell fate commitment.

**Figure supplement 1.** The impact of PRC2 or Kdm6a/b inhibition on H3K27me3, gene expression, and differentiation in WT and *Tert*[-/-] mESCs.

(*Figure 4A*, sample 13). These data were consistent with visual inspection of the ATAC-seq profiles within the *Pou5f1* promoter, which showed a transition from high to low accessible chromatin signal intensity in WT mESCs but not in *Tert*[-/-] mESCs (*Figure 4—figure supplement 1A*). This failure to repress chromatin accessibility was in keeping with the elevated and persistent expression of *Pou5f1*

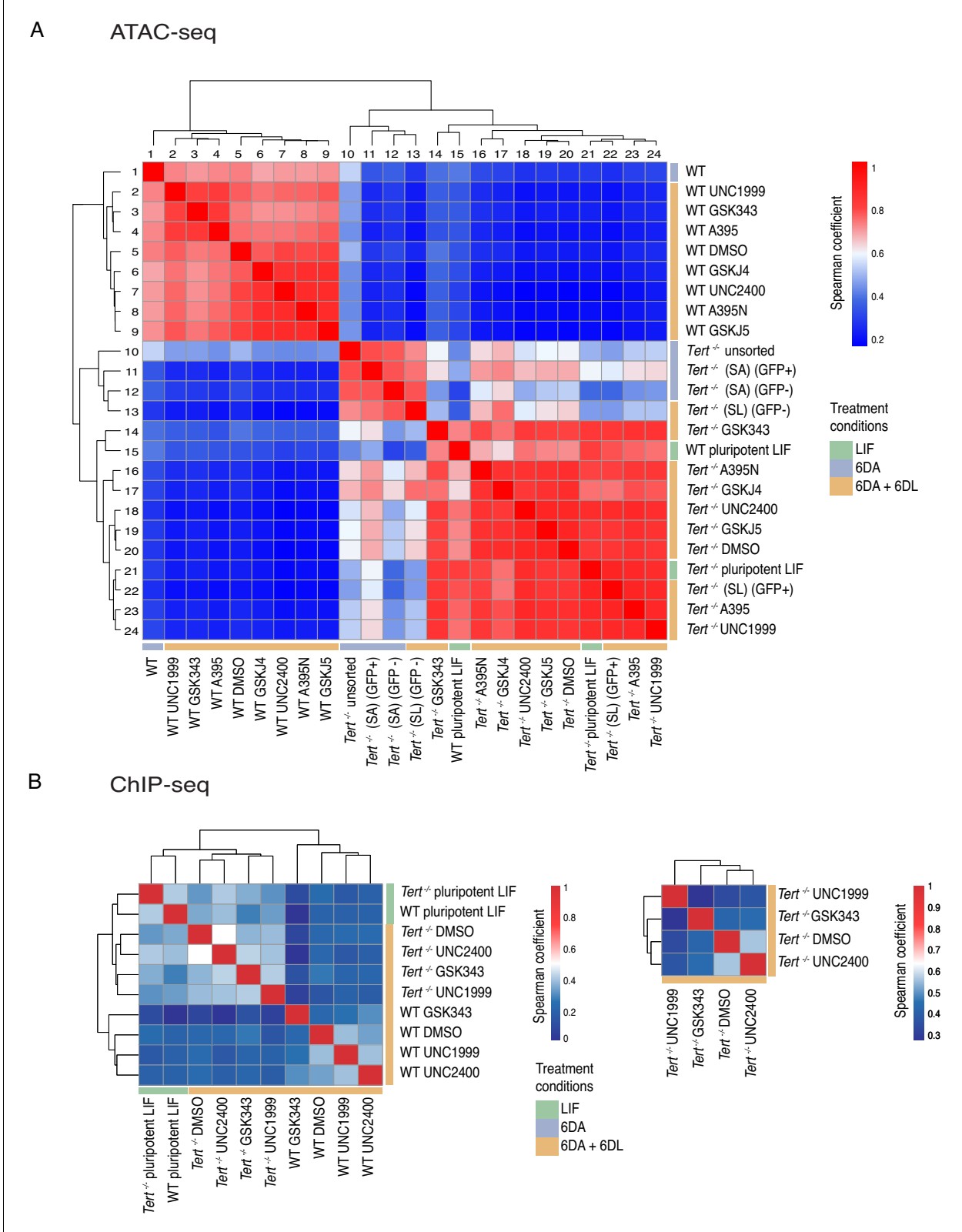

**Figure 4.** Telomere dysfunction affects the chromatin accessibility landscape during mESC differentiation. (**A**) Spearman correlation analysis of called peaks following ATAC-seq analysis of *Tert*[-/-] and WT mESCs assessed throughout differentiation, with or without the addition of PRC2 or Kdm6a/b inhibitors, as indicated (a minimum of 2 biological replicates were analysed per sample; see Materials and methods and Source data files for *Figure 4*). *Figure 4 continued on next page*

*Figure 4 continued*

(B) Spearman correlation analysis of called peaks following H3K27me3 ChiP-seq analysis of the indicated samples. At left, the data are shown for one replicate, and at right, for a subset of samples in which two biological replicates were analysed.

The online version of this article includes the following source data and figure supplement(s) for figure 4:

**Source data 1.** Supplemental information for high throughput sequencing metadata related to ATAC-seq.
**Source data 2.** Supplemental Table 1 related to ATAC-seq data.
**Source data 3.** Supplemental Table 1 related to ChIP-seq data.
**Source data 4.** Supplemental information for high-throughput sequencing metadata related to ChIP-seq.
**Figure supplement 1.** Chromatin accessibility, including at the *Pou5f1* promoter, in WT and *Tert*$^{-/-}$ mESCs during differentiation.

we observed in a subset of *Tert*$^{-/-}$ mESCs. In summary, ATAC-seq analysis bolsters the conclusion that *Tert*$^{-/-}$ mESCs failed to consolidate a differentiated phenotype, and that after LIF re-addition they adopt a distinct chromatin accessible landscape.

To test if the differences in chromatin accessibility were accompanied by differences in H3K27me3 recruitment, we performed chromatin immunoprecipitation using the validated H3K27me3 antibody followed by high throughput sequencing (ChIP-seq) on a selected subset of PRC2 inhibitors that were analyzed by ATAC-seq (*Figure 4B*, *Figure 4—figure supplement 1B*). Consistent with the ATAC-seq results, Spearman analysis of the H3K27me3 ChIP-seq profiles between undifferentiated WT and *Tert*$^{-/-}$ mESCs showed they were similar (Spearman coefficient = 0.4), and possessed little H3K27me3 at the *Pou5f1* promoter (*Figure 4B*, left panel, *Figure 4—figure supplement 1B*). After ATRA and LIF treatment in the presence of the PRC2 inhibitors GSK343 or UNC1999, the WT mESCs clustered separately from *Tert*$^{-/-}$ mESCs, in keeping with the ATAC-seq and transcriptional analysis. Within the *Tert*$^{-/-}$ mESCs, the two active PRC2 inhibitors clustered together (GSK343, UNC1999) and DMSO and the inactive inhibitor UNC2400 formed a different cluster (*Figure 4B*, left and right panels). The assigned ChIP-seq reads obtained from these differentiated mESC populations were insufficient to enable quantitation of chromatin-associated H3K27me3 at specific loci, however visual examination of the *Pou5f1* promoter revealed that differentiated *Tert*$^{-/-}$ mESCs displayed a different pattern of H3K27me3 compared to WT mESCs (*Figure 4—figure supplement 1B*), including subtly increased or broadened peaks, and a few regions with reduced H3K27me3 as we noted previously (region 35642672–35642810) (*Pucci et al., 2013*). Overall, these data support the ATAC-seq results that the chromatin landscape is notably altered in *Tert*$^{-/-}$ mESCs. The apparent compensatory upregulation of H3K27m3 methylation at the *Pou5f1* promoter in *Tert*$^{-/-}$ mESCs may partially explain why *Tert*$^{-/-}$ mESCs are so exquisitely sensitive to PRC2 inhibition compared to WT mESCs (see below).

## Discussion

The differentiation of mESCs, akin to tissue development in vivo, relies on chromatin remodelling. During this process, the precise regulation of DNA methylation and H3K27me3 are critical to the ability of stem cells to adopt a specific state (*Jones and Wang, 2010*; *Margueron and Reinberg, 2011*). Defects in chromatin organization occur during aging and it has been suggested that such changes in the stem cell compartment may potentially alter the differentiation programs and consequently impact cellular and organismal lifespan (*Beerman and Rossi, 2015*; *Fairweather et al., 1987*; *Jung and Pfeifer, 2015*). Despite these major advances, significant challenges remain in the determination of the mechanisms that contribute to age-related tissue decline. The present study revealed a close relationship between two aging hallmarks: telomere shortening and genome-wide epigenetic alterations. We showed that mESCs with critically short telomeres failed to consolidate a differentiated state after exposure to ATRA (*Figure 5*).

Further study is required to define the precise transcriptional and chromatin accessibility alterations in *Tert*$^{-/-}$ mESCs in response to PRC2 or Kdm6a/b inhibition. For example, although these inhibitor dosages had no impact on WT mESC differentiation capacity, we did not transcriptionally profile WT ESCs treated with compounds. Furthermore, although overall perturbations in the chromatin landscape were evident in *Tert*$^{-/-}$ mESCs, further ATAC-seq and ChIP-seq analysis would be needed to quantify the impact of PRC2 or Kdm6a/b inhibition on the chromatin accessibility or the density of H3K27m3 enrichment at specific pluripotency gene promoters or other loci. Despite these important

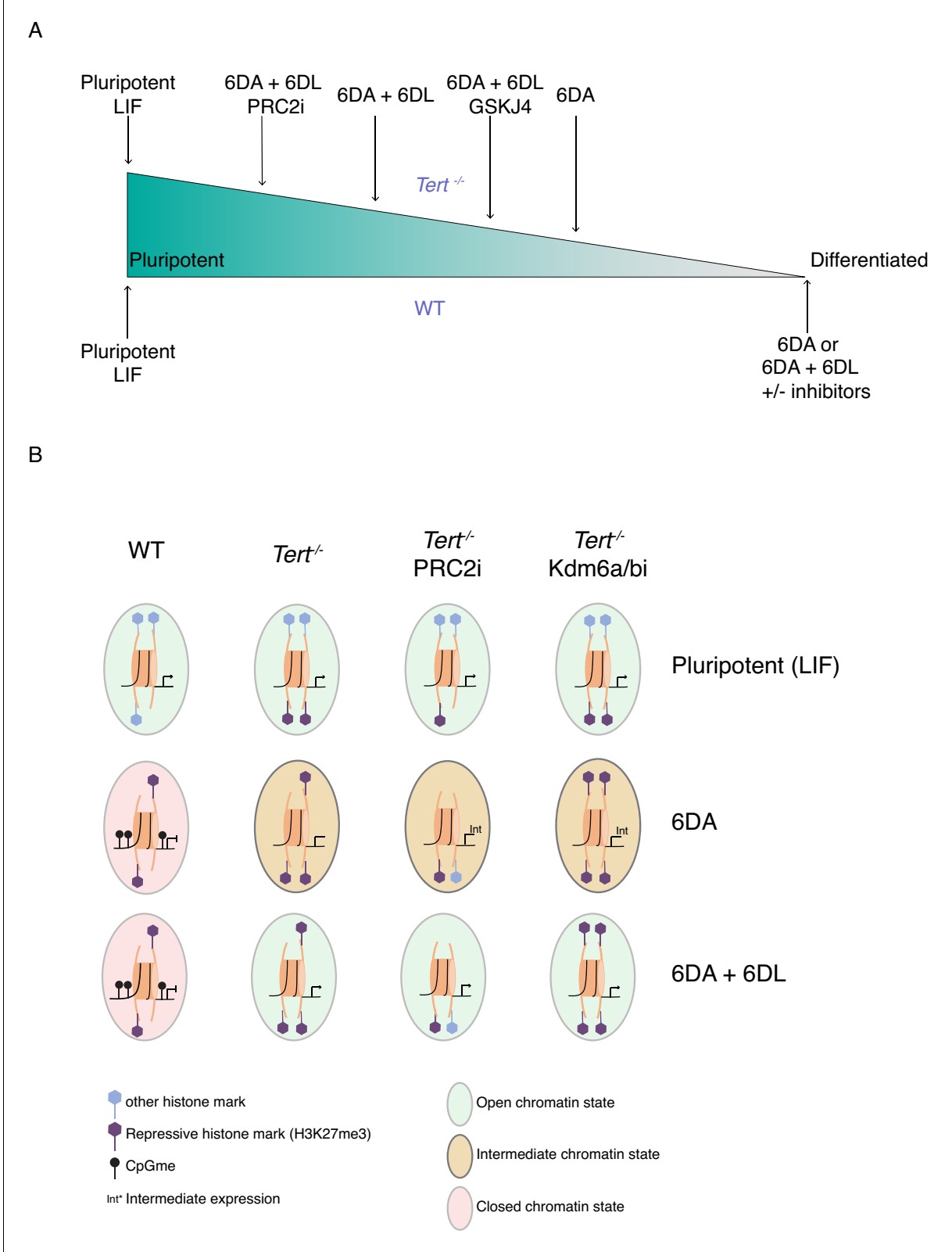

**Figure 5.** The unstable differentiation of mESC with short telomeres is exacerbated by epigenetic remodelling. (A) Schematic of the cell state of WT and *Tert*[-/-] ESCs upon differentiation induction, based on our data. The slope is intended only to illustrate hypothetical relative differences between cell states. (B) Schematic representation of the potential epigenetic landscape(s) at pluripotency gene promoters in WT, *Tert*[-/-], or *Tert*[-/-] mESCs treated with PRC2 or Kdm6a/b inhibitors.

considerations, there was a marked defect in pluripotency marker repression and differentiation commitment that was specific to $Tert^{-/-}$ mESCs upon PRC2 or Kdm6a/b inhibition, both chemically and genetically, and these defects were mirrored by overall alterations in transcription and chromatin accessibility, some of which may reflect this aberrant cell fate commitment.

Given their known roles in the establishment of the heterochromatin state and in agreement with previous studies from other groups, our data also suggest a complex interplay between H3K27me3 and DNA methylation. A study by Hagarman et al. showed that the absence of all three *Dnmt* genes resulted in an increase in the level of H3K27me3 in mESCs (*Hagarman et al., 2013*). This data led to the emergence of two different hypotheses regarding the crosstalk between H3K27me3 and DNA methylation. In one hypothesis, DNA methylation may antagonize H3K27me3 through the exclusion of the PRC2 components from methylated DNA (*Hagarman et al., 2013*). In the alternative model, unmethylated CpG can recruit the PRC2 complex (*Lynch et al., 2012*; *Wachter et al., 2014*) which is known to interact with Dnmts, thereby positively regulating DNA methylation (*Viré et al., 2006*). We have yet to ascertain if the increase in global H3K27me3 levels observed in $Tert^{-/-}$ mESC extracts reflects a significant increase in H3K27me3 recruitment to the promoters of specific loci, however it is interesting to speculate it could be a compensatory mechanism to offset the suboptimal DNA methylation levels when Dnmt levels are limiting. Despite this accumulation of H3K27me3, it is insufficient to overcome the differentiation defect of $Tert^{-/-}$ mESCs, and inhibition of H3K27 methylation led to a further impairment of differentiation (*Figure 5*). In humans, Yu and colleagues suggested a feedback loop between *TERT* and *DNMT3B* transcription. They found that *TERT* depletion in hepatocellular carcinoma cells (HCC) resulted in reduced *DNMT3B* transcription, and they proposed that TERT itself may cooperate with the transcription factor Sp1 to promote *DNMT3B* transcription (*Yu et al., 2018*). This complex interplay between DNA and histone methylation, and how it is affected in $Tert^{-/-}$ mESCs, will be an interesting topic for future investigation.

More generally, this study also examined the impact of PRC2 and Kdm6a/b inhibition on the chromatin accessibility landscape in mESCs after differentiation. To our surprise, the impact of these inhibitors on chromatin accessibility was relatively modest over the 14 days during which WT mESCs were exposed to ATRA followed by LIF. In WT mESCs, this relative resistance to epigenetic perturbation may, in part, be reflected by the relatively low inhibitor concentration and the duration of exposure. Similarly, inhibitor treatment of differentiated $Tert^{-/-}$ mESCs re-exposed to LIF did not dramatically alter the overall chromatin accessibility landscape, perhaps because the landscape was already significantly altered toward a more pluripotent-like state. This result differs from mESCs completely lacking functional PRC2, in which the ability of differentiated somatic cells to be reprogrammed was dramatically impaired (*Pereira et al., 2010*). Thus, the response to PRC2 inhibition is likely dosage-dependent, and under the conditions we tested, inhibition of PRC2 did not significantly impede the differentiation potential of WT mESCs.

Unlike normal cells, the cancer epigenome is exceptionally plastic, particularly within non-coding regions, and is particularly so upon PRC2 inhibition (*Fioravanti et al., 2018*; *Lan et al., 2017*; *Dawson, 2017*; *Stricker et al., 2013*; *Zhou et al., 2016*). PRC2 inhibition also has dramatic consequences for early murine development and cell differentiation in vivo (*Khan et al., 2015*), although the impact on gene expression in adults is more restricted to bivalently controlled, tissue-restricted transcripts (*Jadhav et al., 2016*). In this regard, differentiated $Tert^{-/-}$ mESCs could also be considered more plastic, as they demonstrated a strong propensity to re-express pluripotent genes even in the absence of PRC2 inhibitors, and re-acquired a chromatin landscape reminiscent of pluripotent mESCs. Furthermore, this property does not appear to be specific to retinoic acid-induced ESC differentiation. In a recent study, $Terc^{-/-}$ (telomerase RNA knockout) ESCs with short telomeres exhibited a disruption in PRC2/H3K27me3-mediated repression of Follistatin, *Fsp*, that led to impaired epidermal stem cell specification and differentiation (*Liu et al., 2019*). These results underscore that telomere erosion impairs the epigenetic regulation of cell fate specification in multiple contexts.

Further investigation will ascertain exactly how critically short telomeres elicit these genome-wide perturbations, and whether the epigenetic alterations mirror the altered patterns in methylated DNA observed in older humans or in cancer cells with short telomeres (*Hannum et al., 2013*). Since many cancer cell types and aged tissues possess shorter telomeres, it is possible that the plasticity of these epigenetic modifications may, in part, be affected by telomere integrity. If so, stress responses that promote cellular reprogramming might be linked to eroded telomeres. For example, Mosteiro and colleagues found that senescence promoted in vivo programming of murine iPSCs, although the

relevance of telomere integrity in this process was not directly assessed (*Mosteiro et al., 2018*). In cardiomyocytes stimulated to undergo iPSC reprogramming, there was an inverse correlation between reprogramming potential and telomere length (*Aguado et al., 2017*). In primary human cells undergoing senescence, H3K27me3 loss is correlated with gene expression changes that may drive senescence, and depletion of *EZH2* contributes to the Senescence-Associated Secretory Phenotype (SASP) (*Ito et al., 2018*; *Shah et al., 2013*). SASP itself may promote cellular reprogramming via both cell-autonomous and cell non-autonomous mechanisms (*Milanovic et al., 2018*; *Ritschka et al., 2017*). Thus, one possibility is that short telomeres promote cell fate alterations directly, or indirectly via SASP induction.

In this study, we examined stem cells lacking telomerase, and revealed a genetic interdependency between eroded telomeres and epigenetic alterations required for differentiation. There are other connections between epigenetic modifications and telomeres, even when telomerase is present. Knockdown of the TET enzymes Tet1 and Tet2 has been shown to impact telomere integrity and subtelomeric DNA methylation in addition to playing a key role in cell lineage commitment (*Lu et al., 2014*; *Yang et al., 2016*). Another example is the presence of unstable telomeres in humans harboring a *DNMT3B* deficiency (*Sagie et al., 2017*; *Toubiana et al., 2019*). Mutation of the *TERT* promoter, one of the mostly commonly found non-coding mutations in cancer (*Mularoni et al., 2016*; *ICGC/TCGA Pan-Cancer Analysis of Whole Genomes Consortium, 2020*), leads to telomerase upregulation via allele-specific epigenetic alterations involving PRC2 and DNA methylation (*Leão et al., 2019*; *Lee et al., 2019*; *Stern et al., 2017*). The telomeric transcript TERRA associates with PRC2, and in murine iPSCs lacking *Tp53*, Trf1 depletion results in TERRA-PRC2 recruitment to pluripotency genes (*Marión et al., 2019*; *Montero et al., 2018*). Thus, even in situations where telomeres are not critically eroded, epigenetics impinges on telomeres, and telomeres impinge on epigenetics.

Finally, we are intrigued by the discovery that the differentiation defect in *Tert^-/-* mESCs was modestly rescued with the histone de-methyltransferase inhibitor GSKJ4. Although GSKJ4 is highly selective for Kdm6a/b, it does inhibit other JmjC-domain containing histone demethylases such as *Kdm5b* (*Heinemann et al., 2014*). Our work adds a new twist to the investigation of histone demethylase inhibitors as a potential anticancer therapy (*D'Oto et al., 2016*; *Hoffmann et al., 2012*; *Yang et al., 2019a*; *Yang et al., 2019b*). Notwithstanding the potential deleterious effects of telomere repair in conferring cellular immortality, when combined with stabilization of histone methylation it may prove to be a viable strategy to promote the stable differentiation of cancer cells or the maintenance of tissue homeostasis.

## Materials and methods

Further information and requests for resources and reagents should be directed to and will be fulfilled by the lead contact, Lea Harrington (lea.harrington@umontreal.ca).

### Cell lines

The parental cell line used is E14 murine embryonic stem cells, derived from 129J. The wild-type and *Tert^-/-* ESCs were generated and described in *Liu et al. (2000)*. The lines are genotyped using PCR. The genome was confirmed to be murine based on the alignment of raw HTS data (fastq) against the mm9 dataset. The pluripotency status of the lines prior to differentiation was also confirmed by teratoma formation competency in vivo (data not shown), and expression analysis.

### Cell culture

Murine embryonic stem cells (mESCs) were cultured as described in *Pucci et al. (2013)*. Briefly, mESCs were seeded on 100 mm gelatin-coated plates (VWR, Mont-Royal QC, Canada, CABD353003 and Sigma, Oakville, ON, Canada, cat. # G1890) at a density of $0.9 \times 10^6$ cells per plate and propagated in Glasgow Modified Eagle Medium (GMEM, Gibco, Invitrogen, acquired by Thermo-Fisher, Nepean, ON, Canada, cat. # 11710035) supplemented with 15% v/v fetal bovine serum (FBS; Wisent, St-Bruno, QC, Canada, cat. # 920–040), and 50 units/mL penicillin/streptomycin (Invitrogen, cat. # 15070–063), 0.1 mM non-essential amino acids (Life Technologies, acquired by Thermo-Fisher, Napean, ON, Canada, cat. # 11140050), 0.055 mM 2-mercaptoethanol (Life Technologies, cat. # 21985023) and 1000 U/mL of ESGRO LIF ($1 \times 10^7$ units, Millipore/Sigma, cat. #

ESG1107). Cell cultures were maintained at 37°C with 5% v/v $CO_2$. Cells were split every two days, when they reached an approximate confluency of 80%. All cell lines used in this publication were regularly assessed for mycoplasma contamination and remained mycoplasma-free.

## Generation of *Pou5f1*-GFP mESCs

WT and *Tert*[-/-] mESC lines were transduced (at MOI 25) with a lentivirus expressing pGreenZeo under the *Pou5f1* reporter (*Pou5f1*-GFP; System Biosciences, distributed by Cedarlane, Burlington, ON, Canada, cat. # SR10029VA-1) according to the manufacturer's instructions. The two *Pou5f1*-GFP expressing lines (WT and *Tert*[-/-]) were analyzed at comparable passages beyond 40 (and typically less than passage 68). At these passages, *Tert*[-/-] mESCs exhibit shorter telomeres but cell division, as judged by FACS analysis, was not impaired (*Liu et al., 2000*; *Pucci et al., 2013* and data not shown).

## Differentiation assay

Differentiation analysis was performed as in *Pucci et al. (2013)*. Briefly, after 2 days of growth in LIF-containing media (as above), cells were plated at a density of approximately $0.2 \times 10^6$ cells/100 mm plate on non-gelatin-coated dishes (VWR, cat. # CABD353003), in media without LIF but supplemented with 5 µM all-trans retinoic acid (ATRA, Sigma, R2625-50MG) for 6 days. At day 6, cells were trypsinized using 0.05% w/v trypsin-EDTA 0.5 mM (Invitrogen, cat. # 25300–054), transferred to gelatin-coated plates in GMEM media without ATRA and supplemented with LIF (as above) for 6 days or longer, as indicated. Cells were split to ensure they did not exceed 80% confluency After each step of the differentiation assay, and before reseeding the trypsinized cells, $0.2–0.5 \times 10^6$ cells were used for flow cytometry analysis or $3 \times 10^6$ cells for FACS. The pluripotent state was assessed by flow cytometry, using the *Pou5f1*-GFP reporter gene as a quantitative tool (see below). All experiments were repeated at least three times (n = 3 biological replicates), and typically included n = 3 technical replicates. A biological replicate represents a distinct mESC population (usually analyzed on a different day), and a technical replicate is the same biological replicate processed separately (usually on the same day).

## Treatment of mESCs lines with compounds

Compounds were provided by the Structural Genomic Consortium (SGC) (https://www.thesgc.org) or purchased as a powder and dissolved in DMSO at 20 mM. UNC1999 and GSK343 are small competitive inhibitors of the Ezh2 co-factor S-adenosylmethionine (SAM) and are 1000-fold more selective over other Histone Methyl Transferases (HMT). The major difference between UNC1999 and GSK343 is their inhibition selectivity for Ezh2 (compared to Ezh1), at 22- and 60-fold, respectively (https://www.thesgc.org/chemical-probes/GSK343; https://www.thesgc.org/chemical-probes/UNC1999), (*Konze et al., 2013*; *Verma et al., 2012*). A395 was also employed as an allosteric inhibitor of Eed that antagonizes binding to H3K27me3 and leads to impaired PRC2 function (*He et al., 2017*). UNC2400, a less active compound control for UNC1999 (>1,000 fold lower), and an inactive molecule A395N, structurally similar to A395, were also assessed as controls for the effects of their active analogues. GSKJ4 is an inhibitor of H3K27me3 demethylase and GSKJ5 is a less potent, structurally related molecule (https://www.thesgc.org/chemical-probes/GSKJ1). Compounds were provided by the Structural Genomics Consortium or were purchased, for example: A395, A395N (Sigma, cat. # SML1879-25MG, SML1923-5MG) and GSKJ4, GSK343 (Medchem Express, NY, USA, distributed in Canada by Cedarlane, Burlington, ON, Canada, cat. # HY-15648B, HY-13500). The concentrations chosen for analysis were based on published IC50 values (by the SGC and others; see http://www.thesgc.org/chemical-probes). A similar titration of compound concentrations was tested in mESCs to determine the highest concentration (at or below the IC50) at which no impairment in cell growth was observed after 2 days in LIF-containing media (data not shown).

All compounds were employed at a final concentration of 1 µM, except GSK343 (3 µM). For the differentiation analysis (as described above), compounds were added to LIF-containing media for two days and were added again when the cells were re-plated onto ATRA-containing media without LIF for 6 days. At the midpoint of this 6 day incubation period, the media was replaced with fresh media containing compound or DMSO. After 6 days in ATRA-containing media, the cells were switched to media containing LIF and no ATRA for 6 days, with fresh media and compound or

DMSO added at the midpoint of this incubation period. Except where stated otherwise, three biological replicates of each treatment group were analyzed within an experiment, and additionally included technical replicates.

## Quantitative-Fluorescence in situ hybridization (Q-FISH) analysis to measure relative telomere length

Cell preparation mESCs were seeded to reach a confluency of 60–80% in a 10 cm petri-dish. Cells were treated with 0.2 µg/mL KaryoMAX Colcemid (Gibco, Invitrogen, acquired by Thermo-Fisher, Nepean, ON, Canada, cat. # 15210–040) for 6 hr to synchronize the cells in metaphase.

### Cell swelling and fixation

After collection, cells were centrifuged (5 min 200 x *g*) and the supernatant was removed. 10 mL of 0.075 M KCl was gently added to the cells and incubated for 30 min at 37°C with gentle agitation to keep the cells in suspension. Cells were subjected to centrifugation (as above) and fixed by addition of a solution containing 3:1 methanol:glacial acetic acid) and incubated overnight. For a 10 cm dish, 10 mL of fixative was used.

### Slide preparation

Slides were sprayed with cold water (distilled, deionized) droplets prior to metaphase spread preparation.

### Nuclei preparation for microscopy

After fixation, cells were centrifuged (5 min, 200 x *g*) and the fixative was removed. The pellet was resuspended into 1 mL of fresh fixative and cells were dropped onto the slides. After dropping, slides were immediately washed three times with fixative and incubated for 1 min at 70°C in a humidifier chamber. Slides were dried overnight at room temperature.

### Slide hybridization

Slides were rehydrated with 1X PBS (Phosphate Buffered Saline, 127 mM NaCl, 2.7 mM KCl, 1 mM $Na_2HPO_4$, 1.8 mM $KH_2PO_4$) for 15 min and fixed with 4% v/v formaldehyde/1X PBS for 2 min. Three 1X PBS washes of 5 min each were performed. Slides were treated with 1 mg/mL pepsin in 1X PBS + 0.08 % v/v HCl, washed 2 times with 1X PBS and fixed as described above. Slides were then dehydrated in 70% v/v, 90% v/v and 100 % v/v ethanol (5 min each) followed by air drying. Hybridization to the telomeric probe was performed in the dark using 50 µL per slide of the Cy3 TelC probe (PNA-bio, distributed by Cedarlane, Burlington, ON, Canada, cat. # F1002) in an hybridization buffer (70% v/v ultra-pure deionized formamide, 0.2% w/v BSA, 10 mM Tris-HCl pH 7.4, 0.5 µL/mL of 50 µL PNA TelC-Cy3), incubated in a humidifier chamber at 80°C for 3 min and cooled down overnight at room temperature. The slides were washed twice for 15 min each in wash solution I: 70% v/v pre-deionized formamide (Fisher Scientific, Saint-Laurent, QC, Canada, cat. # 46-505-00ML), 0.1% w/v BSA, 0.01 M Tris-HCl pH 7.4. Slides were washed three times in a 0.075% v/v Tween20/Tris Buffered Saline 1x solution, 5 min per wash and dehydrated as described above. Vectashield with DAPI (Vector Laboratories, distributed by MJS BioLynx, Brockville, ON, Canada, cat. # H-1200) was added to each slide and the coverslip was sealed with nail polish.

### Quantitative fluorescence in situ hybridization microscopy

Standard calibration beads, TetraSpec microspheres 0.1 µM blue/green/orange (Life Technologies, cat. # T7279) were used to ensure that the laser fluctuation did not influence the quality of the analysis. Exposure times were kept constant throughout the image acquisition on the Microscope Zeiss Axio-Imager Z1 using DAPI (excitation, G365 nm; emission, BP 445/50 nm) and Rhodamine (excitation, BP 546/12 nm; emission, BP 575–640 nm) filters. Pictures were then analyzed using the software TFL-telo (https://www.flintbox.com/public/project/502). A minimum of 1587 telomeres were quantified per sample.

## Flow cytometry and cell sorting

Murine ESCs expressing the *Pou5f1*-GFP transgene were sorted at specific times (*Figure 1A*) during the differentiation analysis using the Canto II (Becton Dickinson) analyzer or the ARIA (Becton Dickinson) cell sorter. Cells were harvested using trypsin, washed in PBS and resuspended into 0.2 mL of FACS buffer (PBS 1X, 2% v/v FBS and 0.05 M EDTA). Data were acquired with BD FACSDIVA software and results were analyzed using Flowjo v10. The percentage of GFP+ cells in the population was measured as described (*Zheng and Hu, 2012*). Cells were first gated for FSC/SSC to select the live population. Then doublets were excluded from gating and cells were plotted according to their PE (autofluorescence) and GFP expression. Cells which did not contain the *Pou5f1*-GFP construct were used to establish the gating conditions. No evidence of major cell death was observed during flow cytometry. At least 10 000 events within the GFP+ gate were recorded for each experiment and all the experiments were repeated a minimum of three times (biological replicates) with triplicate technical measurements in each experiment.

## Gamma-irradiation and cell viability analysis

Murine ESCs were seeded for 2 days in LIF-containing media prior to exposure to gamma-irradiation. The irradiation time was calculated to expose the cells to 2 or 5 Gray (Gy). Cells were propagated in LIF-containing media for 2 days and analyzed by propidium iodide staining as follows: after trypsinization, cell pellets were washed with cold PBS two times and stained with 0.5 µg/mL of propidium iodide in 1x binding (10 mM Hepes pH 7.4, 140 mM NaCl and 2.5 mM $CaCl_2$) buffer for 20 min at RT in dark and analyzed by flow cytometry (BD Fortessa).

## Differentiation assay following gamma-irradiation

Murine ESCs were seeded and exposed to gamma-irradiation as previously mentioned in the above paragraph. After 2 days of recovery in LIF-containing media, differentiation was induced.

Protein extraction and western blot analysis mESC pellets were lysed in 1x Laemmli buffer (Bio-Rad, Saint-Laurent, QC, Canada, 161–0747) supplemented with 2-mercaptoethanol (355 mM) according to the manufacturer's instructions. Samples were boiled for 10 min at 95°C and centrifuged at high speed (511258 x *g*) for 20 min (Beckman Coulter, Optima Max XP ultracentrifuge, rotor TLA 120.1). The supernatant was collected and stored at −80°C or used immediately. For detection of H3 (total H3 or H3K27me3), protein extracts were resolved on a 15% w/v Sodium Dodecyl Sulfate Polyacrylamide Gel Electrophoresis (SDS-PAGE) containing a 6% w/v stacking gel and transferred to nitrocellulose membrane specific for small proteins (Bio-Rad, cat. # 1620112) using a wet transfer protocol with 20% v/v methanol Tris glycine-based buffer (25 mM Tris-Cl, 192 mM glycine, pH 8.3). After the transfer, the membrane was incubated in blocking solution (5% w/v milk diluted in Tris-Buffered Saline containing 0.1% v/v Tween 20 - TBST) for 1 hr at room temperature. Mouse anti-H3K27me3 primary antibody (Active Motif, distributed by Cedarlane, Burlington, ON, Canada, cat. # 39155) was added to the blocking solution at a dilution of 1:750, and incubated overnight at 4°C. The membrane was then washed three times in TBST followed by incubation with the secondary antibody VeriBlot for IP Detection Reagent HRP (Abcam, Toronto, ON, Canada, cat. # ab131366) at a dilution of 1:2000 for 1 hr at room temperature. The signal was detected using the ECL SuperSignal West Femto maximum sensitivity substrate (Thermo-Fisher, cat. # 34095) according to manufacturer's instructions, followed by exposure to film (GE Healthcare Life Sciences, distributed by Cedarlane, Burlington, ON, Canada, cat. # 28906839). The membrane was stripped in a prewarmed (50°C) stripping buffer containing 10% w/v SDS, 0.5M Tris HCL pH 6.8% and 0.8% v/v 2-mercaptoethanol for 15 min at 50°C, washed 3 times for 5 min in TBST (http://www.abcam.com/protocols/western-blot-membrane-stripping-for-restaining-protocol), blocked in the same blocking buffer as above, and re-probed with anti-H3 antibody (Abcam, cat. # Ab1791) at a dilution of 1:10,000 for 1 hr at room temperature in the blocking buffer. The membrane was then incubated with secondary antibody and developed as above. For detection of Nanog and Hsp90ab1, the procedure used was as described above with the following changes: the protein extracts were resolved on a 10% w/v SDS-PAGE gel and 6% w/v stacking gel, and the primary antibodies were anti-Nanog (1:1000 dilution, Bethyl Labs, distributed by Cedarlane, Burlington, ON, Canada, cat. # A300-397A) and anti-Hsp90ab1 (1:2000 dilution, VWR, cat. # 10088–510). Figures contain representative western blots from independent experiments that were replicated at least three times.

## RNA isolation

Total RNA was isolated from cell pellets using the miRNeasy Mini Kit (Qiagen, Toronto, ON, Canada, cat. # 217004) and the miRNeasy Micro Kit (Qiagen, cat. # 217084).

## Real-time qPCR array

RNA integrity was assessed using the Bioanalyzer from Agilent according to the manufacturer's instructions. RNA samples were excluded from the analysis if the RNA integrity number was less than 7. Reverse transcription of RNA was performed using the RT$^2$ First Strand Synthesis Kit according to manufacturer's instructions (Qiagen, cat. # 330404). Gene expression analysis was conducted using the PAMM 508ZC RT$^2$ Profiler PCR Array for mouse cell lineage identification (Qiagen, cat. # 330231). This array contains 84 pathway-focused genes (5 of which are pluripotency markers) and five housekeeping genes (*Actb*, *B2m*, *Gapdh*, *Gusb* and *Hsp90ab1*). Controls for mouse genomic DNA contamination, reverse transcription efficiency, and PCR amplification efficiency are included on each array (see Source data files for further information). Quantitative PCR amplification was carried out using the Applied Biosystem StepOnePlus and RT$^2$ SYBR Green Mastermix (Qiagen, cat. # 330523) according to the manufacturer's protocol. Fold-changes in gene expression (after normalization to the five housekeeping genes) were determined using the Qiagen data analysis center at: https://www.qiagen.com/ca/shop/genes-and-pathways/data-analysis-center-overview-page/custom-rt2-pcr-arrays-data-analysis-center/.

## Real-time qPCR for single gene amplification

Reverse transcription of RNA was performed using the RT$^2$ First Strand Synthesis Kit according to manufacturer's instructions (Qiagen, cat. # 330404). Quantitative PCR amplification was carried out using the Applied Biosystem StepOnePlus and the FastStart Universal SYBR Green Master (Roche, cat 04913850001), according to the manufacturer's instructions. Please refer to *Supplementary file 1* for the specific oligonucleotides used for the amplification of specific mRNA species.

## ATAC-seq

The following ATAC-seq protocol was adapted as follows from *Buenrostro et al. (2015)*. As indicated in the source data file to accompany *Figure 4*, a total of 56 samples were analyzed over a total of 3 separate sequencing runs. The sample groups comprise two different genotypes (WT or *Tert*$^{-/-}$) that were isolated by FACS after different steps in the differentiation protocol (i.e. as in the schematic in *Figure 1A*), or after differentiation and LIF re-addition (6 DA + 6 DL) that were untreated or treated with compound, as indicated. A minimum of 2 biological replicates were analyzed for all samples, except two samples for which there were three biological replicates and two samples for which there were three biological replicates and two technical replicates. These additional replicates were included to ensure reproducibility between the three separate occasions on which samples were prepared for sequencing (experimental sets 1–3). Only one sample (a third replicate of KO.6DA_GFP_pos) failed the quality control (# peaks < 20 M), and was excluded from further ATAC-seq analysis. See Source Data files within *Figure 4* for further information.

### Cell thawing and viability measurements

To maximize cell viability, frozen cell pellets were rapidly thawed and then diluted to help remove residual DMSO. Cryotubes were thawed with continuous agitation by hand in a 37°C water bath until only a small piece of ice roughly the size of a grain of rice remained. Cells were transferred by pipette to a 50 mL conical centrifuge tube and 10 mL of pre-warmed 37°C Dulbecco's phosphate buffered saline (PBS; Sigma cat. # 14190–144) was added dropwise with constant gentle swirling of the sample. Samples were then placed on ice and maintained at 4°C for the remainder of the processing time. Cells were pelleted at 500 x *g* for 10 min at 4°C. PBS was removed without disturbing cell pellets and 1 mL of 4°C PBS was then added to resuspend the cells by pipetting. Cell viability was determined via Trypan Blue (Thermo Fisher, cat. # T8154) and counted on a haemocytometer. The equivalent of 50,000 live cells were transferred to a 1.5 mL Eppendorf tube and cells were pelleted by centrifugation at 600 x *g* for 10 min at 4°C.

## Nuclei preparation

ATAC-seq assays were performed essentially as previously described by *Buenrostro et al. (2015)*. All nuclei preparation procedures were performed on ice. After pelleting, PBS was removed by pipette while being careful not to disrupt the cell pellet. Cells were lysed by the addition of 100 µL of cold lysis buffer (10 mM Tris-HCl pH 7.4, 10 mM NaCl, 3 mM $MgCl_2$, 0.1% v/v IGEPAL Ca630 (Sigma, cat. # I8896), and 0.1 % v/v Tween 20 (Thermo-Fisher, cat. # 28320) and then incubated on ice for 5 min. Nuclei were pelleted by centrifugation at 600 x *g* for 10 min. The supernatant was removed carefully to avoid disturbing the nuclei pellet and pelleted nuclei were kept on ice proceeding directly to the transposase step.

## Transposase treatment and DNA purification

Transposase treatment was carried out by the addition of 50 µL of Transposase Master Mix (25 µL Nextera TD Buffer (Illumina, San Diego, CA, USA, cat. # FC-121–1030), 2.5 µL Nextera TDE1 Tn5 Transposase (Illumina cat. # FC-121–1030), 0.5 µL 10% v/v Tween-20 (Thermo-Fisher cat. # 28320), and 22 µL of molecular biology grade water (Sigma, cat. # W4502)) followed by incubation at 37°C for 30 min in a hot block. Post-incubation samples were transferred onto ice, purified with Qiagen MinElute PCR Purification columns (Qiagen, cat. # 28004) and eluted in 15 µL of molecular biology grade water (Sigma, cat. # W4502).

## PCR optimization and amplification

The number of cycles required to amplify the transposed products was optimized using qPCR. An aliquot of 1.2 µL of purified transposed DNA was removed and diluted 1:1 with molecular biology grade $H_2O$ (Sigma, cat. # W4502). Duplicate PCR reactions were carried out (25 µL NEB Next HiFi 2x PCR mix, 6.25 µL of primer Ad1, 6.25 µL of primer Ad2.X (see specific primer details in *Supplementary file 1*), 1 µL of 1:1 diluted transposed DNA and 11.5 µL molecular biology grade H2O (Sigma, cat. # W4502). Samples were amplified on a Bio-Rad CFX96 qPCR thermal cycler with the following protocol: 72°C for 5 min, 98°C for 30 s and then 20 cycles of 98°C for 10 s, 63°C for 30 s and 72°C for 1 min. Post-cycling samples were held at 4°C until proceeding with the next step. Amplification curves were consulted and a cycle number that occurs just prior to saturation was chosen for each sample (typically 10–12 cycles). Once the correct number of cycles for each sample was determined, amplification reactions were prepared (25 µL NEB Next HiFi 2x PCR mix, 6.25 µL of primer Ad1, 6.25 µL of primer Ad2.X, and 12.5 µL transposed DNA). Each sample was amplified using a different Ad2.X primer to allow for sample pooling. PCR was carried out using the same temperature profile as during optimisation but with the determined number of cycles using a Thermo-Fisher Veriti thermal cycler. Post-amplification samples were purified with Qiagen MinElute PCR Purification columns (Qiagen, cat. # 28004) and eluted in 20 µL of molecular biology grade water (Sigma, cat. # W4502).

## Size selection and quality check of libraries

Post-purification, libraries were individually size-selected using a Pippin HT (Sage Science) and 2% w/v agarose gel cassettes (Sage Science). The Pippin HT was set to collect products in the range of 240 to 360 bp. Proper size selection was validated by running a 1 µL aliquot on a BioAnalyzer (Agilent) using a High Sensitivity DNA kit (Agilent, cat. # 5067–4625).

## Assessment of open region enrichment

To determine if the ATAC reactions were successful, an assessment of the overall enrichment of open chromatin regions was assayed by qPCR. Amplification of two canonically open (GAPDH, KAT6B) and one canonically closed region (QML7/8) was performed in duplicate by qPCR on a Bio-Rad CFX96 thermal cycler. qPCR reactions were set up as follows: (5 µL 2x Kapa SYBER FAST Master Mix (Roche cat # 07959389001), 0.5 µL Forward Primer (*Supplementary file 1* for details), 0.5 µL Reverse Primer (*Supplementary file 1* for details), 2 µL molecular biology grade water (Sigma, cat. # W4502). The following amplification protocol was utilized: 72°C for 5 min, 98°C for 30 s and then 40 cycles of 98°C for 10 s, 63°C for 30 s and 72°C for 1 min. Enrichment scores were calculated for each possible open vs closed region ($2(Ct\_Open - Ct\_Closed)$). Scores over 10 were considered to be positively enriched.

## Sequencing

Sequencing libraries were quantified with the Kapa Library Quantification Kit (Roche, distributed by Sigma-Aldrich, cat. # 07960140001). Libraries were normalized to 4 nM and then pooled and denatured with 0.2N NaOH. Libraries were further diluted to a final concentration of 1.4 pM. The pooled libraries were loaded onto an Illumina NextSeq 500 (1.3 mL total) for cluster generation and sequencing. A single-end 75 bp protocol was used and ~60 million reads per sample were targeted.

## Read analysis after ATAC-seq

Fifty-thousand cells were processed from each sample as described above. The resulting libraries were sequenced with 75 bp single-end reads (*Buenrostro et al., 2013*) which were trimmed to 36bps and mapped to mm9 using Bowtie2 (v2.0.5) (http://bowtie-bio.sourceforge.net/bowtie2/index.shtml) with default parameters. Any sample with reads less than 10M and/or peaks less than 20K were excluded from further analysis (there was only one such sample, as noted above). Alignment statistics of all samples and replicates have been listed in *Figure 4—source data 1*. Poor quality reads (MAPQ <= 30, *Zhang et al., 2008*), reads mapping to chrM, chrY and patch chromosomes were removed. Reads were further filtered to remove PCR duplicates using Picard Tool MarkDuplicates (v2.6.0, https://broadinstitute.github.io/picard/). Accessible chromatin regions (peaks) were called using MACS2 v2.0.10. Default parameters were used except for the following: `—keep-dup` all `-B —` `nomodel` `—SPMR` `-q` 0.05. The signal intensity was calculated as the fold enrichment of the signal per million reads in a sample over a modelled local background using the bdgcmp function in MACS2. We then calculated the maximum signal intensity over every peak for conditions being compared using mapBed –o max and quantile normalized them across samples. The average normalized signal intensity across replicates for each condition was then plotted in PCA. All downstream analyses (e.g. Spearman correlation coefficient analysis) were carried out using Bedtools v2.26.0 (https://bedtools.readthedocs.io/en/latest/) and R 3.4.1 (https://www.r-project.org/). Figures were plotted using ggplot2 (https://ggplot2.tidyverse.org/).

## ChIP-seq

### Sample preparation

Chromatin immunoprecipitation methods were adapted from Cardin, Bilodeau and colleagues (*Cardin et al., 2019*). Murine ESCs were harvested, washed with PBS and cross-linked with 1% v/v formaldehyde (methanol-free) for 7 min at room temperature (RT) with agitation. The reaction was quenched by adding 0.125 mM of glycine for 5 min RT. The fixed cells were centrifuged (3 min, 3000 rpm), resuspended in PBS and aliquoted (1 million cells per immunoprecipitation). Aliquots were centrifuged for 10 min 3000 rpm, and flash-frozen with liquid nitrogen. Aliquots were either used immediately or kept at −80°C until immunoprecipitation.

### Preparation of beads

30 µL of the protein A Dynabeads (Thermofisher, #10002D) were used per immunoprecipitation (IP). Using a magnetic rack, beads were washed two times with the ChIP dilution buffer (1% v/v Triton, 10 mM Tris pH 8.0, 150 mM NaCl, 2 mM EDTA) and incubated overnight at 4°C in a low-binding Eppendorf tube on a rotator in the presence of 5 µg of antibody (Active Motif, distributed by Cedarlane, Burlington, ON, Canada, cat. # 39155) in ChIP dilution buffer. Subsequently, beads were washed twice with ChIP dilution buffer and resuspend in 250 µL ChIP dilution buffer and 140 µL of TpS buffer (0.5% v/v Triton, 10 mM Tris pH 8.0, 140 mM NaCl, 1 mM EDTA, 0.5 mM EGTA and protease inhibitors) before immunoprecipitation.

### Chromatin preparation and sonication

Crosslinked cells were thawed, resuspended in TpA buffer (0.25 % v/v Triton, 10 mM Tris pH 8.0, 10 mM EDTA, 0.5 mM EGTA) and incubated on ice for 5 min. Cell nuclei were pelleted by centrifuging for 5 min at 5000 rcf (8,000 rpm) at 4°C, resuspended in TpB buffer (200 mM NaCl, 10 mM Tris pH 8.0, 1 mM EDTA, 0.5 mM EGTA, protease inhibitors) and incubated on ice for 30 min and centrifuged for 5 min at 5000 rcf (8,000 rpm) at 4°C. Finally, the lysed nuclei were resuspended in 140 µL of TpS-SDS buffer (0.5% v/v SDS, 0.5% v/v Triton, 10 mM TRIS pH 8.0, 140 mM NaCl, 1 mM EDTA, 0.5 mM EGTA and protease inhibitors).

Samples were transferred into Covaris tube and sonicated on the Covaris E220 instrument (DC10%, 105W, 200CBP, 2 min) to generate DNA fragments. A small amount (10 µL) of DNA was kept as input.

### Immunoprecipitation

Chromatin preparations were added to the beads and incubated for 4 hr at 4°C on the rotator. Using a magnetic rack, samples were washed subsequently with ice cold low buffer W1 (0.5% v/v NP40, 150 mM KCl, 10 mM Tris pH 8.0, 1 mM EDTA), W2 (0.5 % v/v Triton, 0.1% v/v NaDOC, 100 mM NaCl, 10 mM Tris pH 8.0), W3A (0.5% v/v Triton, 0.1% v/v NaDOC, 400 mM NaCl, 10 mM Tris pH 8.0), W3B (0.5% v/v Triton, 0.1% v/v NaDOC, 500 mM NaCl, 10 mM Tris pH 8.0, W4 (0.5% v/v NP40, 0.5% v/v NaDOC, 250 mM LiCl, 10 mM Tris pH 8.0, 1 mM EDTA) before proceeding to elution.

### Elution and DNA extraction

120 µL of TpE (0.3% w/v SDS, 50 mM Tris pH 8.0, 10 mM EDTA, 0.4 M NaCl) was added to the washed samples and incubated overnight at 65°C with agitation at (200 x g) and DNA was purified by phenol/chloroform.

### Sample quantification and library preparation

DNA samples were quantified using Qubit dsDNA HS Assay kit (ThermoFisher cat # Q32851) to determine concentration of each DNA sample. Library preparations were performed starting with 1 ng of DNA from input DNA or ChIP DNA in a final volume of 10 µL following ThruPlex DNA-seq kit (Takara Bio USA cat # R400676), which consisted of the addition of 3 µL Template Preparation D Master Mix (2 µL Template Preparation D Buffer and 1 µL Template preparation D Enzyme) to each sample, mixed thoroughly by pipetting, and incubated on an Eppendorf Master cycler at 22°C for 25 min, 55°C for 20 min, 4°C hold. Once samples reached 4°C, they were removed from thermal cycler for the addition for 2 µL of Library Synthesis D Master Mix (1 µL Library Synthesis D Buffer, 1 µL Library Synthesis D Enzyme), incubated on thermal cycler at 22°C for 40 min, 4°C hold. 30 µL of library amplification master mix (25 µL Library Amplification D Buffer, 1 µL Library Amplification Enzyme, 3.25 µL Nuclease free water, 0.75 µL 10X SyBr Green I dye 1:1000 dilution) were added to each sample, followed by 5 ul of indexing reagent (SMARTer DNA Unique Dual Index) for a final volume of 50 µL, stored on ice, dark. Ten µL was removed and transferred to a different plate for PCR cycle optimization using. The number of cycles required to amplify the library was optimized using a Bio-Rad CFX96 qPCR thermal cycler with the following program:

| Step 1 | 1 cycle | 72°C | 3 min |
|---|---|---|---|
| Step 2 | 1 cycle | 85°C | 2 min |
| Step 3 | 1 cycle | 98°C | 2 min |
| Step 4 | 4 cycles | 98°C | 20 secs |
| | | 67°C | 20 secs |
| | | 72°C | 40 secs |
| Step 5 | 16 cycles | 98°C | 20 s |
| | | 72°C | 50 s |
| Step 6 | 1 cycle | 4°C | Hold |

The optimal number of cycles at step five were determined from melt curve results (ranging from 8 to 12 cycles). An additional 10 µL of nuclease free water was added to each sample prior to final PCR amplification using Eppendorf Master cycler using sample program with adjusted cycles at step 5. Post amplification samples were purified with 35 µL AmPure XP beads (Beckman Coulter cat # A63881), washed twice with 80% v/v ethanol and eluted in 15 µL of EB buffer (Qiagen cat # 19086), 1 µL was loaded on an Agilent BioAnalyzer (Agilent) using a High Sensitivity DNA kit (Agilent, cat. # 5067–4625) for library size determination.

## Sequencing

One µL of each library were pooled together, quantified by Kapa Library Quantification Kit (Roche cat. # 07960140001). Samples were diluted to 4 nM, denatured with 0.2 N NaOH, and the library pool was further diluted to 12.5 pM prior to loading on an Illumina Miseq Nano kit for cluster and sequencing to determine amount required for deep sequencing. Libraries were pool based on Miseq read counts, diluted to 1.25 nM, denatured with 0.2 N NaOH, neutralized with 400 mM Tris-HCL pH 8 and diluted to a final loading concentration of 250pM. Pooled libraries were loaded onto an Illumina Novaseq6000 sequencer for cluster generation and sequencing to achieve a minimum of 20 million reads per sample using 100 bp single read.

## Read analysis after ChIP-seq

Sequences obtained above were mapped to mm9 using Bowtie2 (v2.0.5) (http://bowtie-bio.source-forge.net/bowtie2/index.shtml) with default parameters. Alignment statistics of all samples and replicates are listed in *Figure 4—source data 3*. Poor quality reads (MAPQ <= 30, *Zhang et al., 2008*), reads mapping to chrM, chrY and patch chromosomes were removed. Reads were further filtered to remove PCR duplicates using Picard Tool MarkDuplicates (v2.6.0, https://broadinstitute.github.io/picard/). H3K27me3 peaks were called using MACS2 v2.0.10 using the following parameters `--keep-dup all -B --broad -q 0.05`. The signal intensity was calculated as the fold enrichment of the signal per million reads in a sample over the input sample. We then calculated the maximum signal intensity over every peak for conditions being compared using mapBed –o max and quantile normalized them across samples. Spearman correlation was then calculated between the average normalized signal intensity across replicates for each condition which was then plotted in heatmap. Analysis downstream of peak calling were carried out using Bedtools v2.26.0 (https://bedtools.readthedocs.io/en/latest/) and R 3.4.1 (https://www.r-project.org/). Figures were plotted using ggplot2 (https://ggplot2.tidyverse.org/).

## Single-guide RNA nucleofection

Eight µg of a synthetic-guide RNA (sgRNA) (Synthego, customized) were mixed with 15 µg of a purified recombinant Hifi Cas9 nuclease V3 (IDT, cat. #262104678), incubated 20 min at RT, and added to 1 million cells in PBS. Nucleofection was performed using the Lonza 4D nucleofector machine, according to the manufacturer instructions. Following nucleofection, cells were seeded and kept in culture for 4 passages before indel quantification and differentiation induction.

## Quantification and statistical analysis

### Cytometry analysis, quantitative real time PCR and western blot quantification

Flow cytometry experiments (*Figures 1*, *2* and *3*) were repeated at least 3 times in triplicate, and values shown represent the mean ± SD. Prism GraphPad was used to perform analysis and statistics. In *Figures 1*, *2* and *3* we performed two-way ANOVA using the standard confidence parameter from the software (alpha = 0.05 and CI 95%) and multiple comparisons. Quantitative real time PCR experiments were repeated in triplicate (except for one n = 2 sample, as indicated in *Figure 1* and *Figure 1—figure supplement 1*) and results were compared using ordinary one-way ANOVA (alpha = 0.05 and CI 95%) (*Figure 1C,D*, *Figure 2—figure supplement 1B*, and *Figure 3—figure supplement 1A,B*). P-values and significance are shown using the GP method in Prism GraphPad, that is $p > 0.5$ = ns (non-significant); * ($p < 0.0332$), ** ($p < 0.0021$), *** ($p < 0.0002$), **** ($p < 0.0001$). Western blots were repeated three times (n = 3) and quantification was performed using Image J (https://imagej.nih.gov/ij/). The ratio of the protein-of-interest to the control protein (as indicated in the figure legends) was analyzed using two-way ANOVA, as described above (see Source Data files for raw data values).

## Real-Time quantitative PCR analysis

Analysis of the Qiagen mouse cell lineage identification qPCR array data was performed following the manufacturer's guidelines. Briefly, following completion of the qPCR run using the Applied Biosystems StepOne Plus cycler, the baseline value was set with the automated baseline option. Using the log view of the amplification plot, the threshold value was chosen within the linear range of the

curve. The threshold value was kept constant across all RT2 profiler runs in the same analysis. A dissociation curve analysis was also performed to ensure PCR specificity. CT values were exported to an Excel spreadsheet for use with the Qiagen Data Analysis Center Web-based software (https://www.qiagen.com/us/shop/genes-and-pathways/data-analysis-center-overview-page/). The lower limit of detection was set to 35 CT. Normalization by average arithmetic mean was performed using CT values of 5 housekeeping genes (*Actb*, *B2m*, *Gapdh*, *Gusb* and *Hsp90ab1*). Fold change in gene expression relative to control sample (eg. WT undifferentiated or WT DMSO) was calculated using the delta delta CT method as described in the manufacturer protocol and using the Web-based software mentioned above (see Source Data files for raw data values).

### Fold change analysis

Statistical analysis of fold change in gene expression (e.g. as in *Figure 2—figure supplement 1B*) was calculated using GraphPad Prism 7. Ordinary two-way ANOVA was performed to compare the mean of each sample with the mean of every other sample. Sidak's multiple comparisons test was applied to correct for multiple comparisons using statistical hypothesis testing, with a single pooled variance. Each P value was adjusted to account for multiple comparisons and the confidence interval was set at 95%.

### Principal component analysis (PCA) plots and heatmaps

Using the fold change values obtained as described above, PCA plots and heatmaps were generated using ClustVis (https://biit.cs.ut.ee/clustvis/) (*Metsalu and Vilo, 2015*). Row centering was performed and unit variance scaling for rows was applied. Principal components were calculated using SVD with imputation method. Principal components with the highest percentage of variance (PC1 and PC2) were used for the PCA plots. For the generation of heatmaps, Euclidean clustering distance was calculated for rows and columns and the linkage criterion (clustering method) was set as complete. Tree ordering for rows and column was set to have the tightest cluster first. Additional details about reagents, primers, kits used in this publication can be found in the *Supplementary file 1*.

## Acknowledgements

The authors thank the Toronto Structural Genomics Consortium (SGC), including Dr. Suzanne Ackloo and Dr. Taylor Mitchell, for advice and providing compounds, and Zhibin Lu (University Health Network) for assistance with ftp data transfers. We thank the IRIC Cytometry facility, including Danièle Gagné, Dr. Gaël Dulude, and Angélique Bellemare Pelletier for technical assistance with FACS, Nadine Mayotte for assistance with the gamma-irradiator, and the IRIC Microscopy facility and Dr. Christian Charbonneau for microscopy training services. We thank present and past members of the lab and colleagues for advice on experiments or analysis, including Dean Betts, Mike Tyers, Yahya Benslimane, Fabio Pucci, Camille Simon, Brian Wilhelm and Thomas Milan, Guy Sauvageau and Bernard Lehnertz (thanked also for supplying the AML cell lines), and E Andrea Mejia Alfaro and Corinne St-Denis for technical assistance. LH acknowledges financial support from the Canadian Institutes for Health Research (Canada) and past support from the Welcome Trust, UK (084637). The SGC is a registered charity (number 1097737) that receives funds from AbbVie, Bayer Pharma AG, Boehringer Ingelheim, Canada Foundation for Innovation, Eshelman Institute for Innovation, Genome Canada through Ontario Genomics Institute (OGI-055), Innovative Medicines Initiative (EU/EFPIA) (ULTRA-DD grant no. 115766), Janssen, Merck KGaA, Darmstadt, Germany, MSD, Novartis Pharma AG, Ontario Ministry of Research, Innovation and Science (MRIS), Pfizer, São Paulo Research Foundation-FAPESP, Takeda, and the Wellcome Trust.

## Additional information

### Funding

| Funder | Grant reference number | Author |
| --- | --- | --- |
| Canadian Institutes of Health Research | 367427 | Lea Harrington |

| Wellcome | 084637 | Lea Harrington |
|---|---|---|
| Ontario Genomics Institute | OGI-055 | Cheryl H Arrowsmith |

The funders had no role in study design, data collection and interpretation, or the decision to submit the work for publication.

### Author contributions
Mélanie Criqui, Conceptualization, Formal analysis, Validation, Investigation, Visualization, Methodology; Aditi Qamra, Data curation, Software, Formal analysis, Validation, Investigation, Visualization, Methodology; Tsz Wai Chu, Conceptualization, Software, Formal analysis, Validation, Investigation, Visualization, Methodology; Monika Sharma, Validation, Investigation, Methodology; Julissa Tsao, Formal analysis, Supervision, Validation, Investigation, Visualization, Writing - review and editing; Danielle A Henry, Validation, Investigation; Dalia Barsyte-Lovejoy, Conceptualization, Methodology, Writing - review and editing; Cheryl H Arrowsmith, Mathieu Lupien, Resources, Supervision, Funding acquisition, Methodology; Neil Winegarden, Resources, Supervision, Methodology, Project administration; Lea Harrington, Conceptualization, Resources, Data curation, Formal analysis, Supervision, Funding acquisition, Methodology, Project administration

### Author ORCIDs
Mathieu Lupien (iD) http://orcid.org/0000-0003-0929-9478
Lea Harrington (iD) https://orcid.org/0000-0002-4977-2744

### Decision letter and Author response
Decision letter https://doi.org/10.7554/eLife.47333.sa1
Author response https://doi.org/10.7554/eLife.47333.sa2

## Additional files

### Supplementary files
• Supplementary file 1. Key resources table. Supplemental information about sequence-based reagents, cells lines, antibodies, chemical compounds, software, algorithms and commercial kits used in this study.

• Transparent reporting form

### Data availability
ATAC-seq and ChIP-seq data has been deposited in GEO under accession number GSE130780 and GSE146322. The Metadata sheet accompanying this deposition is provided in Figure 4 - source data files 2 and 4.

The following datasets were generated:

| Author(s) | Year | Dataset title | Dataset URL | Database and Identifier |
|---|---|---|---|---|
| Criqui M, Qamra A, Chu TW, Sharma M, Henry D, Barsyte D, Arrowsmith CH, Winegarden N, Lupien M, Harrington L | 2020 | Telomere dysfunction cooperates with epigenetic alterations to impair murine embryonic stem cell fate commitment | https://www.ncbi.nlm.nih.gov/geo/query/acc.cgi?acc=GSE130780 | NCBI Gene Expression Omnibus, GSE130780 |
| Criqui M, Qamra A, Chu TW, Sharma M, Henry D, Barsyte D, Arrowsmith CH, Winegarden N, Lupien M, Harrington L | 2020 | Telomere dysfunction cooperates with epigenetic alterations to impair murine embryonic stem cell fate commitment | https://www.ncbi.nlm.nih.gov/geo/query/acc.cgi?acc=GSE146322 | NCBI Gene Expression Omnibus, GSE146322 |

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
