## [Decision Letter]

**Acceptance summary:**

The work reported in this manuscript elaborates details of an intriguing interplay between replicative aging, differentiation and epigenetic histone methylation marks. Critically short telomeres interfere with cell differentiation maintenance mechanisms and the thorough work reported here shows that enzymes that affect histone methylation states are critical contributing factors for these changes. Future work must reveal the mechanistic relation between short telomeres and these genome wide epigenetic markers. Such insights potentially provide exciting new avenues towards an understanding of the differentiation status of cancer cells and, as corollary, of cells in short telomere-induced senescence.

**Decision letter after peer review:**

Thank you for submitting your article "Telomere dysfunction cooperates with epigenetic alterations to impair murine embryonic stem cell fate commitment" for consideration by *eLife*. Your article has been reviewed by three peer reviewers, and the evaluation has been overseen by a Reviewing Editor and Michael Eisen as the Senior Editor. The following individuals involved in review of your submission have agreed to reveal their identity: Jan Karlseder (Reviewer #1); Alexandra Belayew (Reviewer #2); Benoit Laurent (Reviewer #3).

The reviewers have discussed the reviews with one another and the Reviewing Editor has drafted this decision to help you prepare a revised submission.

All three reviewers very much appreciated that you address an important aspect of differentiation stability in embryonic stem cells. The experiments were deemed well thought out and executed and the conclusions therefore, for the most part, very solid and exciting. However, the reviewers also thought that certain results need to be better supported by adding some critical experiments. Upon discussing the manuscript, all reviewers eventually agreed that the following three experiments would be essential in order for the manuscript to go forward:

1) Previously, you used ChIP to report promoter specific changes in H3K27me3 in cell differentiation assays. In the manuscript here, there is a heavy reliance on inhibitors of enzymes that are thought to change the level of this modification genome wide. While a general appraisal of the levels of H3K27me3 in the various conditions is reported by Westerns, it would be important to show the specificity of those changes by evaluating the changes via ChIP-seq on H3K27me3. Such changes could then be related other genome wide analyses of chromatin states (see the ATAC-seq). This experiment, performed on a select choice of conditions, therefore is crucial to bolster the specificity of the effects of enzyme inhibition.

2) The biochemical activities of the PRC2 and JmJD3 enzymes was interfered with using chemical inhibitors. While the reviewers appreciated the use of similar but inactive compounds as controls, they thought that an independent way of inhibition (shRNA mediated knock-down targeting PCR2 core subunits and Jmjd3/KDM6B) would considerably strengthen the robustness of these conclusions.

3) Assuming all the above confirm and strengthen the conclusions, at the end of the day, we still do not know whether the described effect is a telomere-specific effect (due to short telomeres, telomere BFB cycles or telomere damage induced signalling), or is an effect induced by general DNA damage signalling. This differentiation of the source of the signal is extremely important for building a molecular hypothesis and going forward. Therefore, the issue should be settled such that the discussion is relevant to the findings. This could be achieved by exposing wt MEFs to bleomycin or gamma-irradiation to yield a damage signaling similar to the one in the mTERT*^-/-^* MEFs and a few days later, performing a differentiation assay as reported in the paper.

As mentioned above, inclusion and adequate discussion of these experiments is deemed essential for a resubmission and we are looking forward to that.

However, below I also include the other observations/points raised by the reviewers. While it is not essential to address them experimentally, I thought they may be useful when revising the manuscript as certain points could be addressed verbally.

– Could an additional point be addressed in the discussion? The epigenetic age determination methods proposed by Horvath's group and based on changes in methylation of different sets of CpGs has been mentioned in several publications not to be linked to telomere length. Maybe the authors could perform an additional computer analysis of their data to identify where such "epigenetic clock" CpGs are mapped in the chromatin structure landscape, in/out H3K27me3 binding regions? Based on the authors' comment in paragraph two of the Discussion, are these outside of pluripotency genes? Maybe selecting a CpG subgroup with homogeneous location in open chromatin could be more related to telomere length? Or would the authors agree this lack of association could be caused by a telomerase function unrelated to telomere maintenance as suggested in Horvath's publication Lu et al. Nature Communications 2018?

– The authors could address the impact of telomere length versus telomere function. At this time it is unclear, whether the short telomeres, the fused telomeres, the breakage of the fusions, or the deprotected telomeres impact differentiation. This can be addressed by deprotecting long telomeres via TRF2 inhibition, by examining ESC with short telomeres that do not fuse yet, by inhibiting the NHEJ machinery to prevent fusion of critically short telomeres and by inhibiting the signaling from short telomeres via ATM activation.

– Indirect effects of PRC2 and Jmjd6 inhibition will be plentiful, as revealed by the changes in transcription pattern. While I appreciate that it will be very difficult to distinguish between effects directly on telomeres and indirect effects, this should be discussed in more detail.

– ATAC-seq were performed on WT and *mTERT^-/-^* mESCs with PRC2 and Jmjd3/KDM6B inhibitors but in Figure 2D and Figure 2—figure supplement 1F and Figure 3E and Figure 3—figure supplement 1E, the controls on the WT mESCs are missing. The authors should include these controls and modify their conclusions accordingly. Moreover, several data annotated as not shown should be included in the manuscript (see specific comments).

– The authors did an effort of consistency for the annotation of their conditions (6DA, 6DA+6DL) however the PCA figures presented in the main figures are hard to read (not to say illegible) due to the important number of information (especially Figure 1E). The authors should put this type of figures in the supplemental figures and favor the inclusion of the qPCR array heatmap in the main figures. In addition, the clustering the qPCR array data makes difficult to correctly interpret and conclude on the results.

– The differentiation impairment and transcriptional differences are not correlated to the global alteration of chromatin landscape (see specific comment Figure 4A). How the authors reconcile the high global H3K27me3 level observed in *mTert^-/-^* mESCs and the effects of inhibitors (Figure 2B and Figure 3B) with the absence of variation for chromatin accessibility upon inhibitor treatment (Figure 4 and Figure 4—figure supplement 1)? This discrepancy should be addressed by performing H3K27me3 ChIP in presence or absence of PRC2 and Jmjd3/KDM6B inhibitors.

– Figure 1—figure supplement 1A : Is there a difference in telomere length between GFP- and GFP+ cells after 6DA or 6DA + 6DL treatment that could explain the difference of commitment? Also, if the authors are able to recover WT GFP+ cells after 6DA or 6DA + 6DL treatment, it will be great to show telomere dysfunction for this population to strongly support their hypothesis.

– Figure 1—figure supplement 1D :

• The top clustering is very confusing and make the results harder to interpret. The authors should reorganize the heatmap by cell type (e.g. WT mESCs in lane 2, 3, 8 together clearly show that ATRA induces a specific expression pattern that is maintained after re-exposure to LIF) and modify their conclusions accordingly. For example, based on the Figure 1E, the authors mentioned that "the GFP+ subpopulations of mTERT^*-/-*^ mESCs sorted after 6DA or 6DA + 6DL treatment more closely resembled undifferentiated WT mESCs". However, it does not appear to be the case on the qPCR array heatmap (Figure 1—figure supplement 1D lane 10 +11 vs lane 2).

• The GFP- subpopulation of mTERT^-/-^ mESCs sorted after 6DA or 6DA + 6DL treatment (lane 9 + 4) are also able to differentiate and consolidate this differentiation compared to the GFP+ subpopulation (lane 10 + 11). Based on the Figure 1B, 80% of mTert^-/-^ mESCs are GFP- after treatment and therefore able to go to commit towards differentiation – even the expression pattern looks slightly different from the WT (Figure 1—figure supplement 1D lane 4 vs lane 8). What could influence mTert^-/-^ mESCs to commit towards the GFP- more than the GFP+ subpopulation? The authors should at least discuss this point and potentially perform experiments e.g. differences in telomere length or global H3K27me3 level between these subpopulations.

• Based on the clustering, the authors claim that transcriptional profiles did not differ markedly between undifferentiated WT and mTert^-/-^ mESCs (subsection “Compromised telomere integrity alters gene expression profiles”). However by looking more closely, it appears than around 30% of the tested genes show differential expression at basal level (e.g. gene Hand1 to gene Rcvm, gene cd79a to gene Gbx2). The authors should clarify this point and modify their conclusions accordingly.

– Figure 2B and Figure 2—figure supplement 1B : Additional control western blots are required. The authors should include additional histone marks as negative control (e.g. H3K4me1) and as potential crosstalk marks (e.g. H3K27Ac). In addition, the authors should address the PRC2 protein level in WT and mTert^-/-^ mESCs.

– Subsection “Inhibition of PRC2 specifically impairs differentiation in mESCs with

dysfunctional telomeres” : The authors should check the expression of pluripotency markers in WT and mTert^-/-^ mESCs in the LIF media and under the optimized concentration of PRC2 inhibitors.

– In the same section: The authors should include in the manuscript these data referenced as data not shown. How do the authors explain this result? This observation implies that the expression and/or activity of PRC2 is more important after 6DA + 6DL treatment. What is the protein expression level of PRC2 in ESCs maintained in LIF or treated with 6DA compared to ESCs treated with 6DA+6DL?

– Figure 2D and Figure 2—figure supplement 1F the authors should include WT cells treated with PRC2 inhibitors.

– Figure 2E : Why the expression of Nanog by the mTert^-/-^ cells in DMSO conditions are so different between the two western blots while the expression of Hsp90ab1 seems similar? The authors should load all the samples on the same gel to favor a better quantification and comparison between each condition.

– Figure 2—figure supplement 1F: Based on their clustering (Figure 2F), the authors claim that PRC2 inhibitors led to an alteration in the overall gene expression profiles compared with the inactive analogues (subsection “Inhibition of PRC2 specifically impairs differentiation in mESCs with dysfunctional telomeres”). I partially disagree on this statement. We do not see any major differences for GSK343 and UNC1999 vs UNC2400 (left panel Figure 2—figure supplement 1F) while Eed inhibition by A395 vs A395N seems to be more convincing on that aspect. Experiments using shRNA should be performed to clarify the results.

– Figure 3B : Additional control western blots are needed. The authors should include additional histone marks as negative control (e.g. H3K4me1) and as potential crosstalk marks (e.g. H3K27Ac). We do not see any effect of GSKJ5 on H3K27me3 for the WT cells vs mTert^-/-^ mESCs. The authors should address the protein expression level of Jmjd3/KDM6B in WT and mTert^-/-^ mESCs maintained in LIF, treated with 6DA or 6DA+6DL to make sure that Jmjd3/KDM6B protein levels are comparable.

– Figure 3E and Figure 3—figure supplement 1E the authors should include WT cells treated with Jmjd3/KDM6B inhibitors.

– Figure 4A and Figure 4—figure supplement 1: I agree with the authors on this statement "mTert^-/-^ mESCs failed to consolidate a differentiated phenotype", however chemical inhibition (PRC2 or Jmjd3/KDM6B inhibitors) with active or inactive compounds globally does not affect the chromatin accessible landscape for the WT and mTert^-/-^ mESCs. This observation suggests that the global alteration of chromatin landscape does not correlate with the differentiation impairment and transcriptional differences previously described. Example 1: in Figure 2E, Nanog expression is strongly affected by UNC2400 treatment in mTert^-/-^ mESCs when compared to UNC1900 treatment but in Figure 4—figure supplement 1A, there is no difference in chromatin accessibility. Example 2: in Figure 3C, Pou5f1/oct4 decreases after GSKJ4 treatment in mTert^-/-^ mESCs compared to the DMSO or GSKJ5 condition but in Figure 4—figure supplement 1B, there is no difference in chromatin accessibility. The authors should comment on these points and modify their conclusions accordingly. In addition, the authors should perform H3K27me3 ChIP in presence or absence of PRC2 and Jmjd3/KDM6B inhibitors to clarify the interplay between H3K27me3 and stem cell lineage commitment.

---

## [Author Response]

All three reviewers very much appreciated that you address an important aspect of differentiation stability in embryonic stem cells. The experiments were deemed well thought out and executed and the conclusions therefore, for the most part, very solid and exciting. However, the reviewers also thought that certain results need to be better supported by adding some critical experiments. Upon discussing the manuscript, all reviewers eventually agreed that the following three experiments would be essential in order for the manuscript to go forward:1) Previously, you used ChIP to report promoter specific changes in H3K27me3 in cell differentiation assays. In the manuscript here, there is a heavy reliance on inhibitors of enzymes that are thought to change the level of this modification genome wide. While a general appraisal of the levels of H3K27me3 in the various conditions is reported by Westerns, it would be important to show the specificity of those changes by evaluating the changes via ChIP-seq on H3K27me3. Such changes could then be related other genome wide analyses of chromatin states (see the ATAC-seq). This experiment, performed on a select choice of conditions, therefore is crucial to bolster the specificity of the effects of enzyme inhibition.

We have performed the requested ChIP-seq in an extensive set of 14 experimental conditions and controls. First, we show a western blot of an H3K27me3-deficient AML line to demonstrate specificity of the H3K27me3 antibody we used for the ChIP-seq experiments (Figure 2—figure supplement 1, panel A). For the ChIP-seq, rather than quantify peak heights (which would require numerous replicates and only enable comparison of a limited number of samples), we chose to analyze a large cohort of samples and conduct Spearman correlation analysis, as we did for ATAC-seq. We observed the predicted changes in ChIP-seq patterns across the genome in *Tert^-/-^* mESCs compared with WT mESCs. Inspection of specific loci (e.g., *Pou5f1* revealed differences between controls (DMSO) compared with the active PRC2 inhibitors (UNC1999, GSK343). These results support the specificity of the compounds, as the reviewer requested, and the data are discussed in the text, paragraph three of subsection “Telomere dysfunction remodels the chromatin accessibility landscape of differentiated cells toward a pluripotent-like state”. The ChIP-seq data is shown in Figure 4B, Figure 4—figure supplement 1B and the raw data in Figure 4—source data 3 and 4.

2) The biochemical activities of the PRC2 and JmJD3 enzymes was interfered with using chemical inhibitors. While the reviewers appreciated the use of similar but inactive compounds as controls, they thought that an independent way of inhibition (shRNA mediated knock-down targeting PCR2 core subunits and Jmjd3/KDM6B) would considerably strengthen the robustness of these conclusions.

We have performed the requested genetic validation experiments. As discussed in the revised manuscript (paragraph two of subsection “Inhibition of the H3K27me3 erasers Kdm6a/b partially rescues the differentiation impairment of *Tert^-/-^* mESCs.”), we chose to use CRISPR/Cas9-mediated knockouts rather than shRNA knockdown. We found that non-clonal populations containing *Ezh2* knockout cells led to a further exacerbation of the differentiation defect in *Tert^-/-^* mESCs, whereas *Kdm6b* knockout populations led to a partial rescue of the differentiation defect. These genetic results are consistent with the results we obtained with chemical inhibitors of Ezh2 and Kdm6b. This new data is included in Figure 3—figure supplement 1I-L and the corresponding Source data file.

3) Assuming all the above confirm and strengthen the conclusions, at the end of the day, we still do not know whether the described effect is a telomere-specific effect (due to short telomeres, telomere BFB cycles or telomere damage induced signalling), or is an effect induced by general DNA damage signalling. This differentiation of the source of the signal is extremely important for building a molecular hypothesis and going forward. Therefore, the issue should be settled such that the discussion is relevant to the findings. This could be achieved by exposing wt MEFs to bleomycin or gamma-irradiation to yield a damage signaling similar to the one in the mTERT-/- MEFs and a few days later, performing a differentiation assay as reported in the paper.

We have carried out this experiment to discriminate between a telomere-specific effect versus a general DNA damage effect. We tested the effect of DNA damage on the differentiation response of WT mESCs and as further control *Tert^-/-^* mESCs (Figure 1—figure supplement 1E-H). Gamma-irradiation did not affect the WT mESCs, either under pluripotent growth conditions (LIF) or after differentiation. This result argues against a general effect of DNA damage on differentiation commitment under these conditions. In contrast, *Tert^-/-^* mESCs, although slightly less radio-sensitive than WT mESCs, exhibited a further defect in differentiation commitment. These results are interesting and suggest that DNA damage may enhance the impact of a critically eroded telomere, but that the effect is not merely attributable to DNA damage *per se*. They are discussed in the text, paragraph two of subsection “Murine ESCs with critically short telomeres fail to consolidate a differentiated state”.

As mentioned above, inclusion and adequate discussion of these experiments is deemed essential for a resubmission and we are looking forward to that.However, below I also include the other observations/points raised by the reviewers. While it is not essential to address them experimentally, I thought they may be useful when revising the manuscript as certain points could be addressed verbally.– Could an additional point be addressed in the discussion? The epigenetic age determination methods proposed by Horvath's group and based on changes in methylation of different sets of CpGs has been mentioned in several publications not to be linked to telomere length. Maybe the authors could perform an additional computer analysis of their data to identify where such "epigenetic clock" CpGs are mapped in the chromatin structure landscape, in/out H3K27me3 binding regions? Based on the authors' comment in paragraph two of the Discussion, are these outside of pluripotency genes? Maybe selecting a CpG subgroup with homogeneous location in open chromatin could be more related to telomere length? Or would the authors agree this lack of association could be caused by a telomerase function unrelated to telomere maintenance as suggested in Horvath's publication Lu et al. Nature Communications 2018?

The reviewer poses a fascinating question. There are a few points to consider. First, the Horvath clock is based on 353 specific CpGs within the human genome called the CpG clock. These CpGs are located within genomic regions whose expression is regulated during development and differentiation, and some of these regions are regulated by PRC2. Among these sites, some CpGs correlate negatively with chronological aging and others correlate positively. However, the status of any one CpG doesn’t necessarily reflect chronological aging. Performing such a comparison, in our case, would therefore be challenging because it would be difficult to directly map changes in the chromatin landscape between mice and humans. We don’t yet have genome-wide methylome data for *Tert^-/-^* mESCs, and we don’t yet know what the “CpG clock” is in mice with short telomeres (there is a multi-tissue DNA methylation age predictor but only in a wild-type- genetic background with long telomeres, >90 kbp; see Stubbs et al., Genome Biology 2017). Thus, we feel at this juncture it is not possible to address this question.

– The authors could address the impact of telomere length versus telomere function. At this time it is unclear, whether the short telomeres, the fused telomeres, the breakage of the fusions, or the deprotected telomeres impact differentiation. This can be addressed by deprotecting long telomeres via TRF2 inhibition, by examining ESC with short telomeres that do not fuse yet, by inhibiting the NHEJ machinery to prevent fusion of critically short telomeres and by inhibiting the signaling from short telomeres via ATM activation.

This is another interesting question. We previously showed that re-introduction of *Tert* into late passage *Tert^-/-^* mESCs ameliorated the differentiation defect after only 4 passages, which partially rescued critically eroded ends but didn’t significantly elongate telomeres, and that early passage *Tert^-/-^* mESCs with long telomeres do not exhibit a differentiation defect (Pucci et al., 2013). While these data argue that critically short telomeres destabilize differentiation, we have yet to explore the experiments suggested by the reviewer.

– Indirect effects of PRC2 and Jmjd6 inhibition will be plentiful, as revealed by the changes in transcription pattern. While I appreciate that it will be very difficult to distinguish between effects directly on telomeres and indirect effects, this should be discussed in more detail.

We agree, and we now discuss these limitations, Discussion paragraph two.

– ATAC-seq were performed on WT and mTERT-/- mESCs with PRC2 and Jmjd3/KDM6B inhibitors but in Figure 2D and Figure 2—figure supplement 1F and Figure 3E and Figure 3—figure supplement 1E, the controls on the WT mESCs are missing. The authors should include these controls and modify their conclusions accordingly. Moreover, several data annotated as not shown should be included in the manuscript (see specific comments).

In our initial characterization of the inhibitors we tested both WT and *Tert^-/-^* mESCs for their differentiation response, and found no alterations in differentiation commitment. We had shown the data for the 6DA + 6DL samples for WT and *Tert^-/-^* mESCs (Figure 2 B; Figure 3 B) and we now include the data for 2DL and 6DA (Figure 2—figure supplement 1E). Based on this result, we included two controls (WT and *Tert^-/-^* mESCs, 2DL) alongside all the *Tert^-/-^* mESCs (DMSO controls plus inhibitors). It is also likely, as the reviewer points out, that there are alterations in the transcriptional profile of WT mESCs treated with compounds, and we now mention this point in the text (Discussion paragraph two). In the future, we plan to use RNA-seq to address this question in further detail.

– The authors did an effort of consistency for the annotation of their conditions (6DA, 6DA+6DL) however the PCA figures presented in the main figures are hard to read (not to say illegible) due to the important number of information (especially Figure 1E). The authors should put this type of figures in the supplemental figures and favor the inclusion of the qPCR array heatmap in the main figures. In addition, the clustering the qPCR array data makes difficult to correctly interpret and conclude on the results.

We have switched the figure panels as suggested by the reviewer and attempted to clarify the presentation of results, please see subsection “Compromised telomere integrity alters gene expression profiles”, and the summary of Figure changes, attached. We did not adjust the order of the samples in the heatmap in order not to interfere with the information provided via Euclidean clustering.

– The differentiation impairment and transcriptional differences are not correlated to the global alteration of chromatin landscape (see specific comment Figure 4A). How the authors reconcile the high global H3K27me3 level observed in mTert-/- mESCs and the effects of inhibitors (Figure 2B and Figure 3B) with the absence of variation for chromatin accessibility upon inhibitor treatment (Figure 4 and Figure 4—figure supplement 1)? This discrepancy should be addressed by performing H3K27me3 ChIP in presence or absence of PRC2 and Jmjd3/KDM6B inhibitors.

The reviewer raises a good point. We speculate that, since the chromatin is so widely accessible in differentiated *Tert^-/-^* mESCs to begin with, that it might be difficult to quantify additional increases or decreases with the inhibitors, despite the fact they clearly affect pluripotency gene expression. As described above, the ChIP-seq analysis revealed additional H3K27me3 peaks at the promoter of *Pou5f1* and *Nanog* in *Tert^-/-^* mESCs, in addition to the global increase in H3K27me3 we originally reported. However, this enhancement in H3K27me3 was insufficient to repress *Pou5f1* expression, as shown by our transcriptional and ATAC-seq analysis. This result is in keeping with other studies that we cite in the text that is in the absence of sufficient DNA methylation, there is a compensatory increase in H3K27me3. In the case of *Tert^-/-^* mESCs this recruitment is insufficient to restore differentiation stability. We believe the ChIP-seq data may provide some insight into why *Tert^-/-^* mESCs are sensitive to PRC2 inhibitors (paragraph three of subsection “Telomere dysfunction remodels the chromatin accessibility landscape of differentiated cells toward a pluripotent-like state”).

– Figure 1—figure supplement 1A : Is there a difference in telomere length between GFP- and GFP+ cells after 6DA or 6DA + 6DL treatment that could explain the difference of commitment? Also, if the authors are able to recover WT GFP+ cells after 6DA or 6DA + 6DL treatment, it will be great to show telomere dysfunction for this population to strongly support their hypothesis.

We are interested in this question as well. However, we have been hampered in these efforts by the few GFP+ cells we can recover from differentiated WT mESCs which we would need as a control, and the difficulty overall in obtaining enough viable cells after FACS sorting for re-culturing and q-FISH.

– Figure 1—figure supplement 1D :• The top clustering is very confusing and make the results harder to interpret. The authors should reorganize the heatmap by cell type (e.g. WT mESCs in lane 2, 3, 8 together clearly show that ATRA induces a specific expression pattern that is maintained after re-exposure to LIF) and modify their conclusions accordingly. For example, based on the Figure 1E, the authors mentioned that "the GFP+ subpopulations of mTERT-/- mESCs sorted after 6DA or 6DA + 6DL treatment more closely resembled undifferentiated WT mESCs". However, it does not appear to be the case on the qPCR array heatmap (Figure 1—figure supplement 1D lane 10 +11 vs lane 2).

The top labels indicate the Euclidean distance between samples and if we rearranged the order we would lose critical information on the relatedness between samples. We agree that this section needed clarification, and we have attempted to do so in subsection “Compromised telomere integrity alters gene expression profiles”. We also acknowledge that it is merely the relatedness by Euclidean clustering by which we meant “more closely resembled” and that there is indeed a high degree of heterogeneity between the transcriptional profiles.

• The GFP- subpopulation of mTERT-/- mESCs sorted after 6DA or 6DA + 6DL treatment (lane 9 + 4) are also able to differentiate and consolidate this differentiation compared to the GFP+ subpopulation (lane 10 + 11). Based on the Figure 1B, 80% of mTert-/- mESCs are GFP- after treatment and therefore able to go to commit towards differentiation – even the expression pattern looks slightly different from the WT (Figure 1—figure supplement 1D lane 4 vs lane 8). What could influence mTert-/- mESCs to commit towards the GFP- more than the GFP+ subpopulation? The authors should at least discuss this point and potentially perform experiments e.g. differences in telomere length or global H3K27me3 level between these subpopulations.

We have modified the text to state that GFP- *Tert^-/-^* ESCS and WT ESCS are not equivalent, and that we think this difference underscores the aberrant nature of the differentiation program in the presence of short telomeres (subsection “Compromised telomere integrity alters gene expression profiles”).

• Based on the clustering, the authors claim that transcriptional profiles did not differ markedly between undifferentiated WT and mTert-/- mESCs (subsection “Compromised telomere integrity alters gene expression profiles”). However by looking more closely, it appears than around 30% of the tested genes show differential expression at basal level (e.g. gene Hand1 to gene Rcvm, gene cd79a to gene Gbx2). The authors should clarify this point and modify their conclusions accordingly.

We based our conclusion on the profile clustering, but agree there is still heterogeneity in the transcriptional profiles. We have now clarified this issue in the revised text (subsection “Compromised telomere integrity alters gene expression profiles”).

– Figure 2B and Figure 2—figure supplement 1B : Additional control western blots are required. The authors should include additional histone marks as negative control (e.g. H3K4me1) and as potential crosstalk marks (e.g. H3K27Ac). In addition, the authors should address the PRC2 protein level in WT and mTert-/- mESCs.

We previously examined H3K4me3 methylation at the promoters of *Nanog* and *Pou5f1* (Pucci et al., 2013) and found no significant difference. Going forward, we plan to perform ChIP-seq (rather than western blots or ChIP-qPCR) on other histone marks but these are longer-term experiments.

We have assessed the level of mRNA expression of the key PRC2 subunits, and this data is now included in Figure 2—figure supplement 1B. There was no significant difference in the expression patterns between WT and *Tert^-/-^* mESCs during differentiation. We thank the reviewer for suggesting this control.

– Subsection “Inhibition of PRC2 specifically impairs differentiation in mESCs withdysfunctional telomeres” : The authors should check the expression of pluripotency markers in WT and mTert-/- mESCs in the LIF media and under the optimized concentration of PRC2 inhibitors.

The heatmaps show the expression of 5 pluripotency markers in LIF (2DL) for *Tert^-/-^* ESCs Figure 1—figure supplement 1C, and raw data is also included in Source Data file for Figure 1. As mentioned above, we did not perform expression analysis in WT ESCs treated with inhibitors; however we have included additional results after growth in LIF, followed by 6 days in ATRA (Figure 2—figure supplement 1E).

– In the same section: The authors should include in the manuscript these data referenced as data not shown. How do the authors explain this result? This observation implies that the expression and/or activity of PRC2 is more important after 6DA + 6DL treatment. What is the protein expression level of PRC2 in ESCs maintained in LIF or treated with 6DA compared to ESCs treated with 6DA+6DL?

We show that the expression levels of PRC2 subunits doesn’t change appreciably after differentiation. This data is now included as Figure 2—figure supplement 1B. We also believe the ChIP-seq data explains why PRC2 inhibitors have more of an effect on *Tert^-/-^* mESCs as we now explain more fully in the text, paragraph three of subsection “Telomere dysfunction remodels the chromatin accessibility landscape of differentiated cells toward a pluripotent-like state”.

– Figure 2D and Figure 2—figure supplement 1F the authors should include WT cells treated with PRC2 inhibitors.

As described above, we now include the GFP expression data for WT ESCs treated with inhibitors for 2DL and 6DA (Figure 2—figure supplement 1E). Because we found no effect of the inhibitors on the differentiation status of WT ESCs, we did not include all the WT ESCs treated with compounds. In the future, we plan to use RNA-seq to address this question in further detail.

– Figure 2E : Why the expression of Nanog by the mTert-/- cells in DMSO conditions are so different between the two western blots while the expression of Hsp90ab1 seems similar? The authors should load all the samples on the same gel to favor a better quantification and comparison between each condition.

Nanog is consistently elevated in DMSO-treated *Tert^-/-^* mESCs compared to WT ESCs, both by protein and mRNA analysis (Figure 2 C, Figure 2—figure supplement 1 F, G). However, in the top panel in Figure 2 C a shorter exposure time is shown because GSK343 treatment results in a larger increase in Nanog than A395 treatment (i.e. at the same exposure, we would exceed the linear range of the Nanog signal in GSK343). The quantification of the results is shown in Figure 2—figure supplement 1F.

– Figure 2—figure supplement 1F : Based on their clustering (Figure 2F), the authors claim that PRC2 inhibitors led to an alteration in the overall gene expression profiles compared with the inactive analogues (subsection “Inhibition of PRC2 specifically impairs differentiation in mESCs with dysfunctional telomeres”). I partially disagree on this statement. We do not see any major differences for GSK343 and UNC1999 vs UNC2400 (left panel Figure 2—figure supplement 1F) while Eed inhibition by A395 vs A395N seems to be more convincing on that aspect. Experiments using shRNA should be performed to clarify the results.

We concur with the reviewer that the effects of the inhibitors on gene expression profiles is subtle, despite the fact there are clear differences in differentiation commitment that are specific to the active inhibitors. It should also be noted that some of the inactive controls (e.g. UNC2400) are known to retain some residual activity against PRC2. To test these findings genetically, we now include CRISPR-mediated knockout experiments (paragraph two of subsection “Inhibition of the H3K27me3 erasers Kdm6a/b partially rescues the differentiation impairment of *Tert^-/-^* mESCs”).

– Figure 3B : Additional control western blots are needed. The authors should include additional histone marks as negative control (e.g. H3K4me1) and as potential crosstalk marks (e.g. H3K27Ac). We do not see any effect of GSKJ5 on H3K27me3 for the WT cells vs mTert-/- mESCs. The authors should address the protein expression level of Jmjd3/KDM6B in WT and mTert-/- mESCs maintained in LIF, treated with 6DA or 6DA+6DL to make sure that Jmjd3/KDM6B protein levels are comparable.

As mentioned above, we plan to address this potential crosstalk using genome-wide ChIP-seq approaches but believe this exceeds the scope of the present study. Regarding the second question, we now include expression data (mRNA instead of protein) and show that *Kdm6b* mRNA levels did not differ significantly between WT and *Tert^-/-^* mESCs (Figure 3—figure supplement 1A, B). We also show H3K37me3 levels in cells in which we disrupted *Kdm6b* in a heterogeneous, non-clonal knockout population (Figure 3—figure supplement 1I-L) and noted a modest (but not statistically significant) increase in H3K27me3.

– Figure 3E and Figure 3—figure supplement 1E the authors should include WT cells treated with Jmjd3/KDM6B inhibitors.

We now include the results of the differentiation assay as noted above (Figure 2—figure supplement 1E), but we have not performed transcriptional analysis using the Qiagen qPCR array.

– Figure 4A and Figure 4—figure supplement 1: I agree with the authors on this statement "mTert-/- mESCs failed to consolidate a differentiated phenotype", however chemical inhibition (PRC2 or Jmjd3/KDM6B inhibitors) with active or inactive compounds globally does not affect the chromatin accessible landscape for the WT and mTert-/- mESCs. This observation suggests that the global alteration of chromatin landscape does not correlate with the differentiation impairment and transcriptional differences previously described. Example 1: in Figure 2E, Nanog expression is strongly affected by UNC2400 treatment in mTert-/- mESCs when compared to UNC1900 treatment but in Figure 4—figure supplement 1A, there is no difference in chromatin accessibility. Example 2: in Figure 3C, Pou5f1/oct4 decreases after GSKJ4 treatment in mTert-/- mESCs compared to the DMSO or GSKJ5 condition but in Figure 4—figure supplement 1B, there is no difference in chromatin accessibility. The authors should comment on these points and modify their conclusions accordingly. In addition, the authors should perform H3K27me3 ChIP in presence or absence of PRC2 and Jmjd3/KDM6B inhibitors to clarify the interplay between H3K27me3 and stem cell lineage commitment.

We performed ATAC-seq on 72 samples. Although further replicates may have allowed quantification of peak heights we do not believe this would alter our conclusions. We agree that the effects with compounds GSKJ4 and GSKJ5 were modest, but emphasize that we chose relatively low concentrations of these inhibitors purposefully, i.e. to enhance any potential differences in differentiation between WT and *Tert^-/-^* mESCs. The new results we obtained via CRISPR-mediated disruption of *Ezh2* and *Kdm6b* (Figure 3—figure supplement 1I-L) corroborate our conclusions.